# Chemical genetics and proteome-wide site mapping reveal cysteine MARylation by PARP-7 on immune-relevant protein targets

Kelsie M Rodriguez[1], Sara C Buch-Larsen[2], Ilsa T Kirby[1†], Ivan Rodriguez Siordia[1], David Hutin[3], Marit Rasmussen[4], Denis M Grant[3], Larry L David[1], Jason Matthews[3,4], Michael L Nielsen[2], Michael S Cohen[1]*

[1]Department of Chemical Physiology and Biochemistry, Oregon Health & Science University, Portland, United States; [2]Proteomics program, Novo Nordisk Foundation Center for Protein Research, Faculty of Health and Medical Sciences, University of Copenhagen, Copenhagen, Denmark; [3]Department of Pharmacology and Toxicology, University of Toronto, Toronto, Canada; [4]Department of Nutrition, Institute of Basic Medical Sciences, University of Oslo, Oslo, Norway

*For correspondence:
cohenmic@ohsu.edu

Present address: †Department of Cellular and Molecular Pharmacology, University of California, San Francisco, San Francisco, United States

Competing interests: The authors declare that no competing interests exist.

**Abstract** Poly(ADP-ribose) polymerase 7 (PARP-7) has emerged as a critically important member of a large enzyme family that catalyzes ADP-ribosylation in mammalian cells. PARP-7 is a critical regulator of the innate immune response. What remains unclear is the mechanism by which PARP-7 regulates this process, namely because the protein targets of PARP-7 mono-ADP-ribosylation (MARylation) are largely unknown. Here, we combine chemical genetics, proximity labeling, and proteome-wide amino acid ADP-ribosylation site profiling for identifying the direct targets and sites of PARP-7-mediated MARylation in a cellular context. We found that the inactive PARP family member, PARP-13—a critical regulator of the antiviral innate immune response—is a major target of PARP-7. PARP-13 is preferentially MARylated on cysteine residues in its RNA binding zinc finger domain. Proteome-wide ADP-ribosylation analysis reveals cysteine as a major MARylation acceptor of PARP-7. This study provides insight into PARP-7 targeting and MARylation site preference.

## Introduction

Poly-ADP-ribose-polymerases (PARPs) are a family of 17 proteins, many of which have emerged as critical regulators of cytokine signaling and innate immunity (*Fehr et al., 2020*). Most PARP family members catalyze the transfer of a single unit of ADP-ribose from nicotinamide adenine dinucleotide (NAD⁺) to amino acids on protein targets, a reversible post-translational modification (PTM) known as mono-ADP-ribosylation (MARylation) (*Daugherty et al., 2014*). Unlike other well-studied PTMs, such as phosphorylation or ubiquitylation, how PARP-mediated MARylation regulates protein function is still poorly understood.

Multiple independent lines of evidence point to PARP-7 playing a critical role in the innate immune signaling pathway, particularly as a negative regulator of the type I interferon antiviral response. Knockout of PARP-7 enhances nucleic acid sensor agonist- or virus-induced interferon-beta (IFN-β) expression in various cell types (*Yamada et al., 2016*; *Kozaki et al., 2017*). Given the antiviral functions of IFN-β, increases in IFN-β are expected to suppress viral replication. Consistent with this notion, viral titers after infection with several single-strand RNA viruses, including influenza, vesicular stomatitis virus, encephalomyocarditis virus, and mouse hepatitis virus (a murine

coronavirus), are lower in PARP-7 knockout or knockdown cells compared to wild-type cells (*Yamada et al., 2016*; *Grunewald et al., 2020*). These results show that PARP-7 facilitates viral replication, in part, by shutting down IFN-β production. How PARP-7-mediated MARylation regulates IFN-β production, and perhaps other aspects of innate immune signaling, is unknown. This is largely because few direct MARylation targets of PARP-7 are known.

A unique feature of MARylation compared to other PTMs is that chemically diverse amino acids can act as ADP-ribose (ADPr) acceptors (*Cohen and Chang, 2018*). Historically, the chemical nature of the amino acid-MAR bonds in proteins was examined by chemical stability studies (*Payne et al., 1985*; *Cervantes-Laurean et al., 1993*; *Hsia et al., 1985*; *Krantz and Lee, 1976*; *McDonald and Moss, 1994*). This led to the notion that the major ADPr acceptor residues are the acidic amino acids, glutamate (Glu) and aspartate (Asp), as well as the basic amino acid arginine (Arg). More recently, however, mass spectrometry-based proteomics has revealed that in addition to Glu/Asp and Arg, serine (Ser), tyrosine (Tyr), histidine (His), and cysteine (Cys) can also act as ADPr acceptors (*Buch-larsen, 2020*; *Larsen et al., 2018*; *Leslie Pedrioli et al., 2018*; *Palazzo et al., 2018*; *Leidecker et al., 2016*). While many residues can act as ADPr acceptors, a major outstanding question in the field is whether or not individual PARP family members demonstrate selectivity toward ADP-ribosylation of specific residues in protein targets. Identifying the MARylated residues in protein targets is essential for understanding how ADPr modifications regulate protein function.

In this study, we combine chemical genetics, proximity labeling, and proteome-wide amino acid ADPr site profiling for identifying the direct targets and sites of PARP-7-mediated MARylation. Our study reveals that PARP-7 MARylates proteins in innate immune signaling and viral regulation. Additionally, we show that Cys MARylation in cells is more stable than Glu/Asp MARylation, suggesting that the site of MARylation in protein targets governs signal duration.

## Results

### Optimization of a chemical genetic strategy for identifying the direct targets of PARP-7

Identifying the protein targets of PARP-7 is a critical first step toward unraveling its function in cells. In previous work we have developed an engineered enzyme—modified substrate strategy for identifying relevant targets of individual PARP family members (*Carter-O'Connell et al., 2014*; *Carter-O'Connell et al., 2016*; *Carter-O'Connell and Cohen, 2015*; *Carter-O'Connell et al., 2018*). This chemical genetic (CG) method uses engineered PARPs and an orthogonal $NAD^+$ analogue that contains a clickable handle (e.g. an alkyne) that is located at the N-6 position of the adenine ring (5-Bn-6-a-$NAD^+$). We showed that 5-Bn-6-a-$NAD^+$ is an excellent substrate for engineered PARPs in which a hydrophobic amino acid (isoleucine or leucine) at the floor position (so named because it sits at the floor of the nicotinamide subsite in the $NAD^+$ binding pocket) is mutated to a glycine. Importantly, 5-Bn-6-a-$NAD^+$ is not used by wild-type (WT) PARPs and in this way is orthogonal to native $NAD^+$. While 5-Bn-6-a-$NAD^+$ is a very good substrate for many floor position engineered PARPs (e.g. L1782G, LG-PARP-14), we found that it is a poor substrate for several other floor position PARP mutants, including PARP-7. We therefore sought a new orthogonal $NAD^+$ analogue that would serve as an efficient substrate for floor position engineered PARP-7, I631G PARP-7 (IG-PARP-7) (*Figure 1a*).

We focused our efforts on changing the position of the clickable handle. Recently, Pascal and colleagues reported a crystal structure of a non-hydrolyzable analogue of $NAD^+$, benzamide adenine dinucleotide (BAD), bound to PARP-1 (*Langelier et al., 2018*). This is the first crystal structure of a PARP bound to an $NAD^+$ biomimetic and it shows how the adenine ring of $NAD^+$ binds in the $NAD^+$-binding pocket. Based on this structure, it is evident that the C-2 position of the adenine ring of BAD is more solvent exposed than the N-6 position of the adenine ring (*Figure 1b*). Indeed, work from Marx and colleagues showed that C-2 modified $NAD^+$ analogues were better PARP1 substrates than N-6 modified $NAD^+$ analogues (*Wallrodt et al., 2017*). Given the high degree of conservation in the $NAD^+$-binding pocket between PARP family members, we hypothesized that placing the clickable handle at the C-2 rather the N-6 position of 5-Bn-$NAD^+$ would yield a better substrate for IG-PARP-7. We therefore designed and synthesized 5-Bn-2-e-$NAD^+$, which has an ethynyl group at the

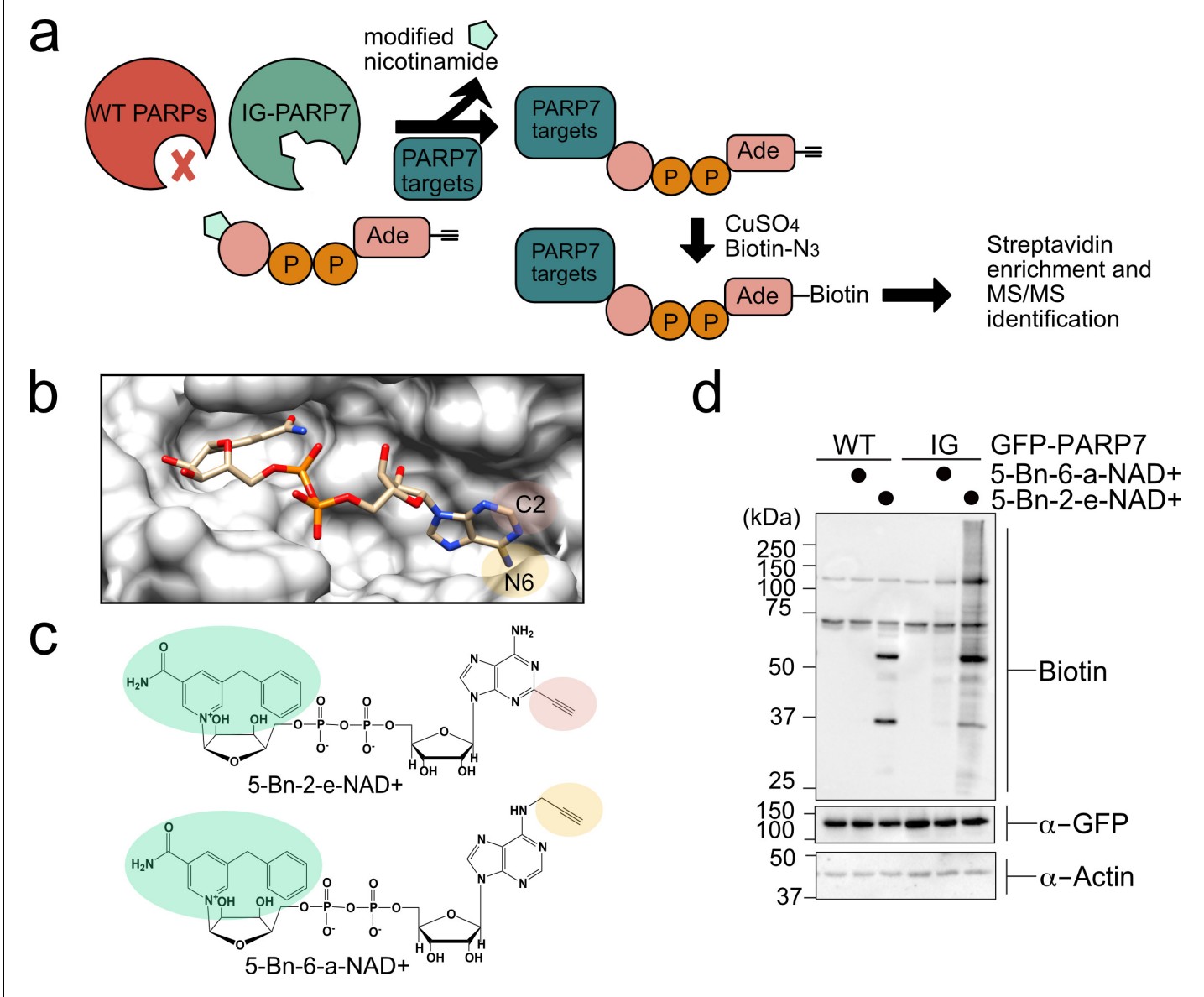

**Figure 1.** A chemical genetic strategy for identifying the direct MARylation targets of PARP-7. (**a**) Schematic of chemical genetic approach for identifying direct targets of PARP-7. (**b**) Crystal structure of human PARP1 bound to the non-hydrolyzable NAD[+] analog benzamide adenine dinucleotide (BAD). PDB: 6bhv. The structure shows that the C-2 position of BAD is pointed out of the ADP-ribose pocket. (**c**) Chemical structure of the clickable, orthogonal NAD[+] analogs. 5-Bn-2-e-NAD[+] is the optimized substrate for the floor position engineered PARP-7, IG-PARP-7. 5-Bn-6-a-NAD[+] was used in our previous studies by other floor position engineered PARPs (e.g. PARP14). (**d**) 5-Bn-2-e-NAD[+] is an efficient substrate for IG-PARP-7. HEK293T cells were transfected with either GFP-WT-PARP-7 (control) or GFP-IG-PARP-7. Cell lysates were prepared and incubated with either 5-Bn-2-e-NAD[+] or 5-Bn-6-a-NAD[+] (100 µM). Following copper-catalyzed click chemistry with biotin-azide, proteins were resolved by SDS-PAGE and biotinylated proteins were detected by Western blot detection using Streptavidin-HRP. Biotinylated proteins represent direct MARylation targets of PARP-7. The online version of this article includes the following figure supplement(s) for figure 1:

**Figure supplement 1.** Enrichment of PARP-7 MARylation targets and labeling of PARP-12 IG targets using a chemical genetic approach.

C-2 position for Cu(I)-catalyzed Huisgen chemistry with an azide reporter (*Figure 1c* and Methods Supplement).

Compared to 5-Bn-6-a-NAD[+], we found that 5-Bn-2-e-NAD[+] is a much better substrate for GFP-IG-PARP-7 as evidenced by the greater labeling (biotin signal following click conjugation to biotin-azide) of GFP-tagged IG-PARP-7 (GFP-IG-PARP-7) (auto-MARylation) as well as the labeling of many other discrete bands (trans-MARylation) in HEK 293 T cell lysates (*Figure 1d*). Importantly, we did

not detect labeling of GFP-WT-PARP-7, indicating that 5-Bn-2-e-NAD$^+$ is not a substrate for GFP-WT-PARP-7 (*Figure 1d*).

We next sought to identify the targets of PARP-7 using GFP-IG-PARP-7 and its optimized substrate, 5-Bn-2-e-NAD$^+$. Following incubation of HEK 293T lysates with 5-Bn-2-e-NAD$^+$ and subsequent conjugation with biotin-azide, we enriched biotinylated proteins (e.g. MARylation targets) using NeutrAvidin agarose (*Figure 1—figure supplement 1a*). We then proteolyzed enriched proteins and eluted peptides were subjected to tandem mass spectrometry (LC-MS/MS). We identified a total of 250 direct targets of PARP-7, many of which are RNA binding and RNA regulatory proteins (*Supplementary file 1*).

## Identification of the PARP-7 interactome using BioID proximity labeling

To obtain a more holistic view of PARP-7 function in cells we used a proximity labeling approach—commonly referred to as BioID—for identifying intracellular PARP-7 interactors (*Figure 2—figure supplement 1a*). In the BioID approach, a protein of interest is fused to a promiscuous biotin ligase (BirA*) (*Roux et al., 2012*; *Roux et al., 2013*). Upon addition of biotin to cultured cells, BirA* uses biotin as a substrate to generate a lysine-reactive adenylate-biotin that reacts with proteins proximal to the fusion protein. In this way, intracellular interactors can be identified. We generated a WT PARP-7 chimeric construct in which a Myc-tagged promiscuous biotin ligase (BirA*) was fused to the N-terminus of PARP-7 (Myc-BirA*-WT-PARP-7). We observed robust biotinylation of proteins across the full molecular weight range in HEK 293 T cells expressing Myc-BirA*-WT-PARP-7, but not GFP-WT-PARP-7 (*Figure 2—figure supplement 1b*). Biotinylated proteins were enriched using NeutrAvidin agarose (*Figure 2—figure supplement 1b*). In the Myc-BirA*-PARP-7 NeutrAvidin pulldown sample we found an enrichment of auto-MARylated Myc-BirA*-PARP-7 as well as trans-MARylated proteins, demonstrating that Myc-BirA*-PARP-7 is catalytically active and that MARylated targets are enriched (*Figure 2—figure supplement 1b*). Enriched proteins were proteolyzed, and eluted peptides were subjected to LC-MS/MS. We identified a total of 189 interacting proteins of PARP-7 (*Supplementary file 1*). We found that a subset of these PARP-7 interactors overlapped with targets identified using our CG approach (*Figure 2a*), suggesting that these interactors could be intracellular MARylation targets of PARP-7.

## Gene ontology enrichment analysis of PARP-7 targets

To gain insight into the cellular functions of PARP-7, we performed gene ontology (GO) analysis using both the direct MARylation targets (CG approach) and the intracellular PARP-7 interactors (BioID approach). We performed GO biological process term enrichment using PANTHER and visualized terms using ReviGO (*Figure 2b*). We found a significant enrichment of terms related to viral processes, RNA regulation, and the innate immune response (*Figure 2b*, *Supplementary file 1*). The enrichment of these terms is consistent with the known role of PARP-7 in the regulation of the innate immune response during viral infection (*Yamada et al., 2016*; *Kozaki et al., 2017*; *Grunewald et al., 2020*).

We also performed protein-protein interaction (PPI) analysis using the list of MARylation targets and intracellular interactors of PARP-7 using the Metascape tool (*Zhou et al., 2019*). To identify densely connected PPI network components, the Molecular Complex Detection (MCODE) algorithm was applied to the Uniprot IDs of all the proteins within our CG and BioID lists. These protein networks show clusters of interactions most abundantly related to mRNA translational silencing, mRNA splicing and metabolism, protein folding and localization to Cajal bodies (*Figure 2c*, *Supplementary file 1*). Taken together, these analyses support the notion that PARP-7 regulates various aspects of RNA regulation, including regulation of RNA localization, RNA splicing, and mRNA translation.

## Validation of PARP-13 as a target of PARP-7 in cells

We next sought to validate candidate proteins as MARylation targets of PARP-7 in intact cells. We focused our attention on PARP-13 (also known as ZAP and ZC3HAV1) as it was a top candidate in both proteomics datasets (*Supplementary file 1*) and because it is a well-known RNA binding protein (*Todorova et al., 2015*; *Guo et al., 2004*). PARP-13 is unique among PARP family members for two main reasons: i. it exists as two major isoforms, the full-length protein known as PARP-13.1

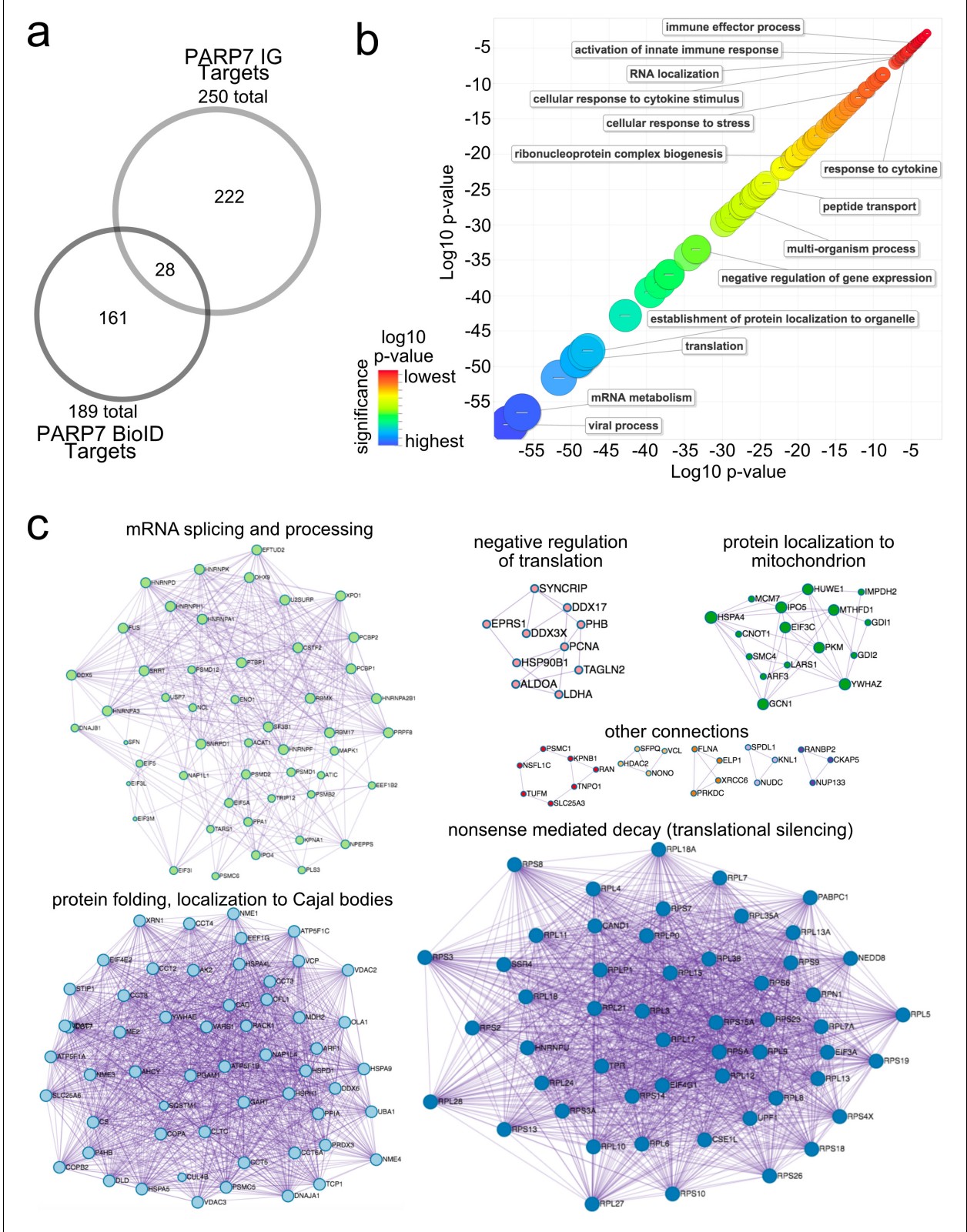

**Figure 2.** Gene ontology analysis of PARP-7 MARylation targets and interactors reveals roles in innate immune signaling and RNA regulation. (**a**) Venn diagram comparing the 250 total PARP-7 targets identified by LC-MS/MS runs in comparison to the 189 PARP-7 interacting proteins identified using Myc-BirA–PARP-7. Data shown are the additive combined targets of two biological replicates for each type of strategy. (**b**) Scatterplot depicting enriched gene ontology (GO) terms attached to the 439 total PARP-7 MARylation targets (chemical genetics) and PARP-7 interactors (BioID proximity

*Figure 2 continued on next page*

*Figure 2 continued*

labeling). GO term enrichment was performed using the PANTHER toolkit. Significantly enriched GO terms (p<0.05) were condensed using ReviGO and similar terms were nested based on similarity. Terms are organized by −log10(p-value). Significance ranges from −58.2 (most significant) to −2.8 (least significant). Selected terms are indicated. (c) 'Protein-protein Interaction Enrichment Analysis' of the 439 total PARP-7 MARylation targets and PARP-7 interactors. Images were created using Metascape.org. The Molecular Complex Detection (MCODE) algorithm has been applied to identify densely connected network components. The MCODE networks identified for individual gene lists have been gathered and are shown here.

The online version of this article includes the following figure supplement(s) for figure 2:

**Figure supplement 1.** A proximity labeling approach identifies intracellular interactors of PARP-7.

(ZAPL) and a truncated variant known as PARP-13.2 (ZAPS) that is devoid of the catalytic domain, and ii. despite having the catalytic domain, PARP-13.1 is catalytically inactive. PARP-13.1 is constitutively expressed in most cells and PARP-13.2 is induced by interferons (e.g. IFN-β) as well as viral infection (*Schwerk et al., 2019*). In HEK 293 T cells, PARP-13.1 and PARP-13.2 are expressed, but PARP-13.1 is the major isoform (*Li et al., 2019*). Indeed, we confirmed that PARP-13.1 is the major endogenous isoform that interacts with Myc-BirA*-PARP-7 in HEK 293 T cells (*Figure 3a*). Similar to PARP-7, both isoforms of PARP-13 are important regulators of innate immunity and the response to viral infection. This functional connection motivated us to determine if both PARP-13.1 and PARP-13.2 are intracellular targets of PARP-7.

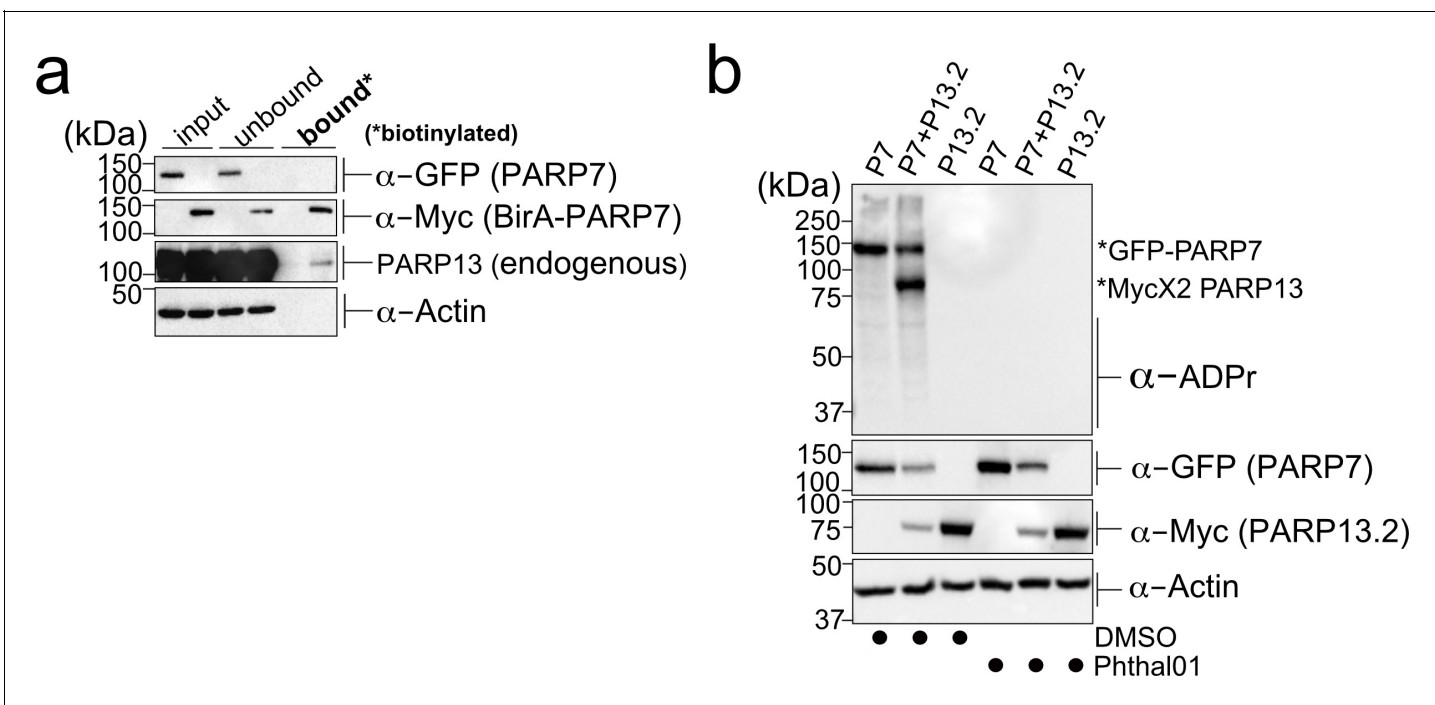

**Figure 3.** PARP-13 interacts with and is MARylated by PARP-7 in cells. (a) PARP-7 interacts with endogenous PARP-13 (constitutive isoform PARP-13.1) in cells as shown by proximity labeling. HEK 293 T cells were transfected with Myc-BirA*-PARP-7. Biotin (50 μM) was added to induce biotinylation of Myc-BirA*-PARP-7 interactors. Biotinylated proteins were enriched using Neutravidin agarose. Proteins were resolved by SDS-PAGE and were detected by Western blot using antibodies against GFP, Myc, PARP-13, and actin. Representative figure of data collected over three biological replicates. (b) PARP-7 MARylates PARP-13.2 in cells. GFP-PARP-7 and MycX2-PARP-13.2 were co-expressed in HEK293T cells. Cells were treated overnight either with DMSO or with Phthal01 (1 μM). Proteins were resolved by SDS-PAGE and were detected by Western blot using antibodies against ADP-ribose (ADPr), GFP, Myc, and actin. Representative figure of data collected over three biological replicates.

The online version of this article includes the following source data and figure supplement(s) for figure 3:

**Source data 1.** In vitro IC$_{50}$ Data for Phthal01 against PARP family members.

**Figure supplement 1.** PARP-13 is a MARylation target of PARP-7.

**Figure supplement 2.** Phthal01 inhibits PARP-7, and at much higher concentrations, PARP-10-mediated MARylation in cells.

To determine if PARP-13.1 and PARP-13.2 are MARylated by PARP-7 in cells, we co-expressed GFP-PARP-7 with MycX2- or HA-tagged PARP-13.1 or PARP-13.2. We found that both PARP-13.1 and PARP-13.2 are MARylated by GFP-PARP-7 (*Figure 3b*, *Figure 3—figure supplement 1a*). Myc-BirA*-PARP-7 MARylated both PARP-13.1 and PARP-13.2 to a similar extent as GFP-PARP-7, indicating that the tag on PARP-7 does not impact PARP-13 MARylation (*Figure 3—figure supplement 1a*). In the absence of PARP-7, we do not detect MARylation of PARP-13.1 or PARP-13.2 (*Figure 3b*, *Figure 3—figure supplement 1a*). Because the similar molecular weights of GFP-PARP-7 and PARP-13.1 made it challenging to distinguish trans-MARylation of PARP-13.1 from auto-MARylation of PARP-7, we focused all of our subsequent studies on PARP-13.2.

We next asked if the catalytic activity of PARP-7 is required for the MARylation of PARP-13.2. For these experiments, we took a pharmacological approach. Recently a phthalazinone-based piperazine compound, AZ12629495, was reported in the patent literature (*Limited and Park, 2009*). AZ12629495 was shown to inhibit PARP-7 as well as PARP-1 and PARP-2 with nanomolar potency (*Limited and Park, 2009*). This was confirmed in another study, but family-wide analysis was not investigated (*Lu et al., 2019*). We synthesized a close analog of AZ12629495, referred to here as Phthal01 (Methods Supplement), and profiled it across the PARP family using an in vitro plate assay for screening PARP inhibitors developed in our lab (*Kirby et al., 2018*).We found that Phthal01 is most potent against PARP-7 ($IC_{50}$ = 14 nM), followed closely by PARP1 ($IC_{50}$ = 21 nM), and PARP2 ($IC_{50}$ = 28 nM) (*Figure 3—source data 1*). Phthal01 is at least 13-fold selective for PARP-7 versus the other PARP family members besides PARP-1 and PARP-2 (*Figure 3—source data 1*). We then evaluated the effects of Phthal01 against PARP-7 in HEK 293 T cells by western blot analysis using an ADPr-specific antibody. We found that Phthal01 dose dependently inhibits auto-MARylation of PARP-7 in GFP-PARP-7 expressing HEK 293 T cells ($EC_{50}$ ~60 nM) (*Figure 3—figure supplement 2*). At much higher concentrations, Phthal01 inhibits auto-MARylation of PARP-10 ($EC_{50}$ ~3 μM) (*Figure 3—figure supplement 2*). Phthal01 (1 μM), but not the PARP1/2-selective inhibitor veliparib (1 μM), completely inhibits both auto-MARylation of PARP-7 and PARP-7-mediated trans-MARylation of PARP-13.2 (*Figure 4—figure supplement 1f*). Taken together, these results show that the that catalytic activity of PARP-7 is required for PARP-13.2 MARylation in cells.

We sought to further validate PARP-13.2 as a direct MARylation target of PARP-7 by performing in vitro MARylation assays with recombinantly expressed PARP-7 and PARP-13.2. In the presence of a physiological concentration of $NAD^+$ (100 μM), GST-PARP-7 trans-MARylated $His_6$-SUMO-PARP-13.2 in addition to itself (auto-MARylation) in a time-dependent manner (*Figure 3—figure supplement 1b*). Despite strong auto-MARylation, $His_6$-SUMO-PARP-10 poorly trans-MARylated $His_6$-SUMO-PARP-13.2 under similar conditions (*Figure 3—figure supplement 1b*). These data further demonstrate that PARP-13.2 is a *bona fide* MARylation target of PARP-7.

Having demonstrated that PARP-13.2 is a MARylation target of PARP-7 we next asked if PARP-7 catalytic activity is required for its interaction with PARP-13.2. Using recombinantly expressed GST-PARP-7 and $His_6$-SUMO-PARP-13.2 we performed co-immunoprecipitation experiments under the following conditions: No $NAD^+$, 100 μM $NAD^+$, or 100 μM $NAD^+$ + Phthal01 (*Figure 3—figure supplement 1c*). GST-PARP-7 co-immunoprecipitated with $His_6$-SUMO-PARP-13.2 under all conditions, demonstrating that the interaction between PARP-7 and PARP-13.2 is independent of PARP-7 catalytic activity.

## Cys residues are the major sites of PARP-7-mediated MARylation in PARP-13

We next sought to identify the amino acid ADPr acceptors in PARP-13 that are MARylated by PARP-7 using chemical stability studies. $NH_2OH$ cleaves the ester bond between Glu/Asp and ADPr whereas $HgCl_2$ cleaves the thioether bond between Cys and ADPr (*Figure 4a*; *Hsia et al., 1985*). We co-expressed HA-PARP-13.2 with either GFP-PARP-7 or GFP-PARP-10 in HEK 293 T cells and evaluated MARylation by Western blot analysis using an ADPr-specific antibody. In lysates from HEK 293 T cells expressing GFP-PARP-7 and HA-tagged PARP-13.2, we found that $HgCl_2$ (2 mM) reduces substantially (~90%) the ADPr signal on HA-PARP-13.2, whereas neutral $NH_2OH$ (0.4 M) only partially reduces (~30%) this signal (*Figure 4b*). In contrast, in lysates from HEK 293 T cells expressing GFP-PARP-10 and HA-tagged PARP-13.2, we found that neutral $NH_2OH$ reduces substantially (~90%) the ADPr-signal on PARP-10 whereas $HgCl_2$ has no effect (*Figure 4b*). The latter result is consistent with previous studies from our lab and others showing

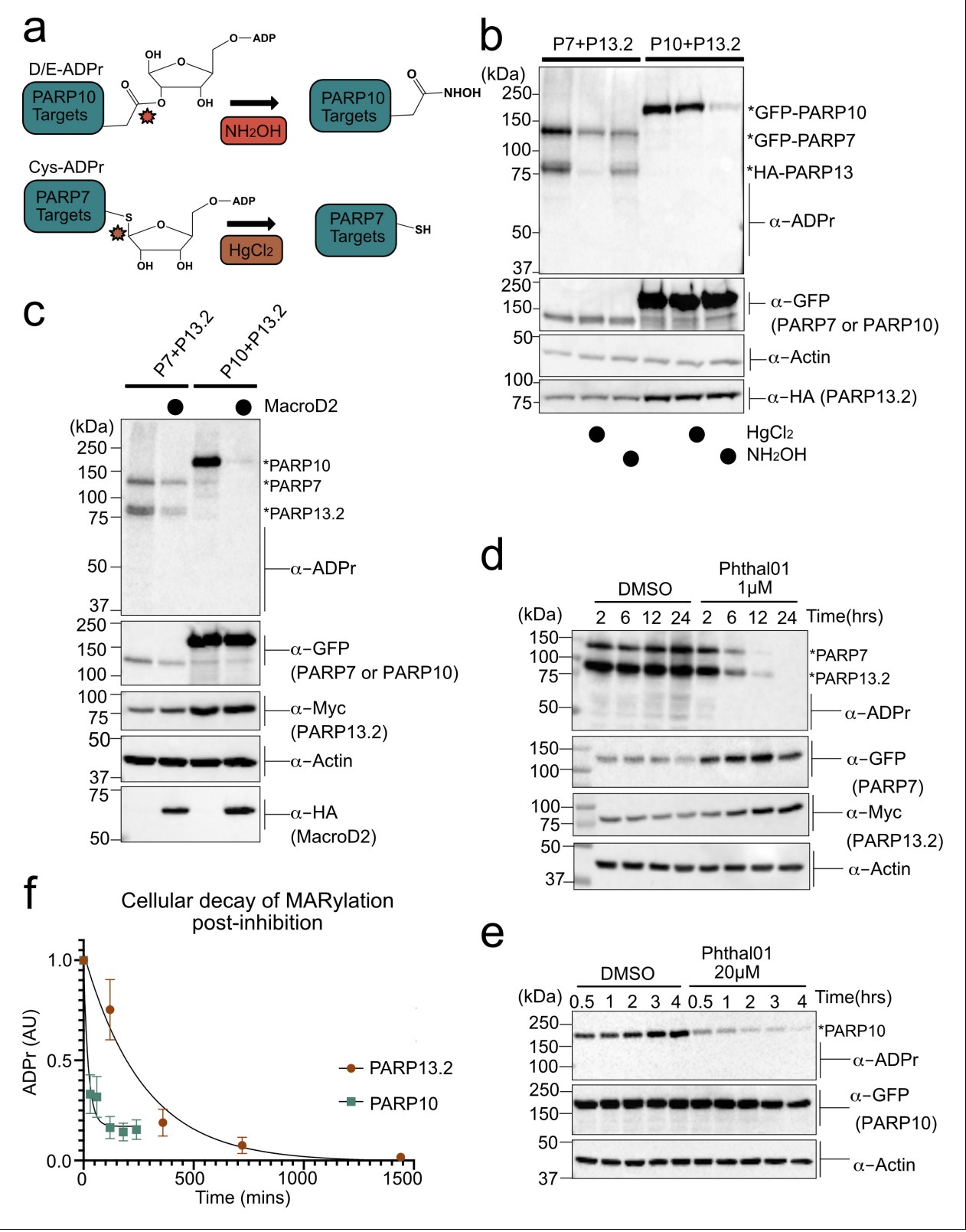

**Figure 4.** PARP-13 is MARylated by PARP-7 predominately on Cys residues and Cys MARylation is more stable than Asp/Glu MARylation in cells. (a) Schematic of chemical stability studies for analyzing the amino acid-ADPr linkage. Neutral hydroxylamine (NH$_2$OH) removes ADPr from acid residues to generate a hydroxamic acid, while mercuric chloride (HgCl$_2$) removes ADPr from Cys residues. (b) PARP-7 MARylates predominately on Cys residues whereas PARP-10 MARylates predominately on Glu/Asp residues. GFP-PARP-7/HA-PARP-13.2 or GFP-PARP-10 alone were expressed in HEK 293 T

*Figure 4 continued on next page*

Figure 4 continued

cells. Lysates were prepared and treated with either water control, HgCl$_2$ (2 mM), or NH$_2$OH (0.4 M). Following treatment, proteins were precipitated with ice cold methanol, and resolved by SDS-PAGE and were detected by Western blot using antibodies against ADPr, GFP, HA, and actin. Representative figure of data collected over three biological replicates. (c) The Glu/Asp selective ADPr hydrolase MacroD2 efficiently removes ADPr from PARP-10, but only partially removes ADPr from PARP-7 and PARP-13. GFP-PARP-7/MycX2-PARP-13.2 or GFP-PARP-10 was co-expressed with either HA-MacroD2 or mCherry (negative control) in HEK 293 T cells. Proteins were resolved by SDS-PAGE and were detected by Western blot using antibodies against ADPr, GFP, Myc, and actin. Representative figure of data collected over three biological replicates. (d) Time course of Cys MARylation in cells. GFP-PARP-7 and MycX2-PARP-13.2 were co-expressed in HEK 293 T cells. Cells were treated with either DMSO or Phthal01 (1 μM) to stop PARP-7 MARylation. Cells were harvested at indicated time points and proteins were resolved by SDS-PAGE and were detected by Western blot using antibodies against ADPr, GFP, Myc, and actin. Representative figure of data collected over three biological replicates. (e) Time course of Glu/Asp MARylation in cells. GFP-PARP-10 was expressed in HEK 293 T cells. Cells were treated with either DMSO or Phthal01 (20 μM) to stop PARP-10 MARylation. Cells were harvested at indicated time points and proteins were resolved by SDS-PAGE and were detected by Western blot using antibodies against ADPr, GFP, and actin. Representative figure of data collected over five biological replicates. (f) Quantification of replicate data from (d) and (e). ADPr signal is normalized to either GFP-PARP-10 or MycX2-PARP-13.2. Data are ± SEM.

The online version of this article includes the following figure supplement(s) for figure 4:

**Figure supplement 1.** PARP-13.2 Cys MARylation is more stable than PARP-10 Asp/Glu MARylation in cells.
**Figure supplement 2.** Inhibition of PARP-7 catalytic activity stabilizes PARP-7 protein levels.

that PARP-10 is auto-MARylated predominately on Glu/Asp (*Morgan and Cohen, 2015*; *Kleine et al., 2008*). Despite the strong auto-MARylation activity of PARP-10 in cells, we did not detect trans-MARylation of PARP-13.2 when these two constructs were co-expressed (*Figure 4b*). Treatment with either neutral NH$_2$OH or HgCl$_2$ equally reduces PARP-7 auto-MARylation suggesting it may be modified on both cysteine and acidic residues (*Figure 4b*). Although it is formally possible that another PARP modifies PARP-7 on acidic amino acids, the observation that PARP-7 MARylation is completely inhibited by Phthal01 (1 μM), but not veliparib (1 μM), suggests that these acid amino acids ADPr acceptor residues are auto-MARylation sites (*Figure 4b*, *Figure 4—figure supplement 1f*). Taken together, these results show PARP-13.2 is a selective target of PARP-7 that it is MARylated predominately on Cys residues.

To further evaluate the ADPr acceptor residues in PARP-13, we took an enzymatic approach. Previous studies demonstrate that the macrodomain containing enzyme macroD2 removes ADPr attached to Glu/Asp, thereby reversing Glu/Asp-directed MARylation (*Glowacki et al., 2009*; *Jankevicius et al., 2013*; *Rosenthal et al., 2013*). While it is not known if macroD2 removes ADPr attached to Cys, it does not remove ADPr attached to Ser (*Fontana et al., 2017*). We co-expressed HA-macroD2 with either GFP-PARP-7 and MycX2-PARP-13.2 or GFP-PARP-10 and MycX2-PARP-13.2 in HEK 293 T cells and evaluated MARylation by western blot analysis using an ADPr-specific antibody. We found that while macroD2 efficiently reverses auto-MARylation of PARP-10, macroD2 only partially reverses both auto-MARylation of PARP-7 and trans-MARylation of PARP-13.2 (*Figure 4c*). These results are consistent with the chemical stability studies and further support the notion that PARP-7 predominately MARylates Cys residues in PARP-13.2.

## Intracellular MARylation stability is dependent on the chemical nature of the amino acid-ADPr bond

For many PTMs involved in cell signaling (e.g. phosphorylation), the rate of reversal is the major determinant of signal duration. It is unknown if disparate amino acid-ADPr bonds exhibit different cellular stabilities, which could impact the signal duration of MARylation. We measured the decay of PARP-7-mediated PARP-13 MARylation, which occurs predominately on Cys, and PARP-10 auto-MARylation, which occurs predominately on Glu/Asp. We used Phthal01 to stop the forward MARylation reaction: 1 μM for PARP-7 and 20 μM for PARP-10. We then harvested cells at different time points to measure the decay of PARP-7-mediated MARylation of PARP-13.2 and auto-MARylation of PARP-10 by western blot using an ADPr-specific antibody. We found that the half-life (t$_{1/2}$) of ADPr in PARP-10 is 15 min whereas the t$_{1/2}$ of ADPr in PARP-13.2 is 189 min (*Figure 4d–f*, *Figure 4—figure supplement 1*). Importantly, the PARP-1/2 inhibitor veliparib does not lead to the decay of PARP-7-mediated MARylation of PARP-13.2 (*Figure 4—figure supplement 1f*). The longer t$_{1/2}$ of MARylated PARP-13.2 versus MARylated PARP-10 suggests that the Cys-ADPr bond is more stable in HEK 293 T cells than the Glu/Asp-ADPr bond.

We next determined if the longer $t_{1/2}$ for Cys-ADPr compared to Glu/Asp-ADPr was due to increased chemical or enzymatic stability. Lysates from HEK293T cells expressing PARP-10 or PARP-7 together with PARP-13.2 were treated with 2% SDS and heated to 95°C to completely denature proteins; in this way, the inherent chemical stability of the Cys- versus Glu/Asp-ADPr bind could be assessed. We observed no change in the Cys- or Glu/Asp-ADPr signal over time (up to 24 hr) (*Figure 4—figure supplement 2a*).

This result shows that Cys-ADPr is more stable than Glu/Asp-ADPr in HEK 293 T cells.

## Inhibition of PARP-7 catalytic activity increases its protein levels

The HEK 293 T cell time course experiments revealed that Phthal01 increased the protein levels of ectopically expressed GFP-PARP-7 and MycX2-PARP-13.2 (*Figure 4*, *Figure 4—figure supplement 1*). To test if endogenous PARP-7 and PARP-13 protein levels are similarly affected by inhibition of PARP-7 catalytic activity, we treated A549 cells, which express endogenously both PARP-7 and PARP-13, with Phthal01. PARP-13 was detected using two commercially available PARP-13 antibodies, whereas PARP-7 was detected using a recently developed PARP-7 antibody (manuscript in preparation, J. Matthews). The specificity of the PARP-7 antibody was validated using *PARP-7*[+/+] and *PARP-7*[-/-] mouse embryonic fibroblasts (*Figure 4—figure supplement 2b*; *MacPherson et al., 2013*). We found that endogenous PARP-7 protein levels increase substantially upon treatment with Phthal01; however, the levels of endogenous PARP-13.1 did not change (*Figure 4—figure supplement 2c*). PARP-13.1 is the major endogenously expressed isoform under basal conditions. These results suggest that while endogenous PARP-7 levels are regulated by its catalytic activity, endogenous PARP-13.1 levels are not regulated by PARP-7 catalytic activity under these conditions.

## PARP-7 predominately MARylates Cys residues in the CCCH zinc finger domains of PARP-13

The chemical stability and enzymatic removal studies point to Cys residues as the predominate amino acids in PARP-13 that are MARylated by PARP-7. However, we sought direct evidence for PARP-7-mediated Cys MARylation in PARP-13, and perhaps other targets. Recently, Nielsen and colleagues established an unbiased MS platform for confidently identifying and localizing the amino acid ADPr acceptors in proteins on a proteome-wide scale (*Hendriks et al., 2019*; *Larsen et al., 2018*). This strategy involves the enrichment of ADP-ribosylated peptides (generated post trypsin digest) using the macrodomain Af1521, which binds with high affinity to ADPr-modified peptides. Enriched peptides containing ADPr are subjected to MS-based analysis using electron-transfer higher energy collisional dissociation (EThcD), a fragmentation technique that can localize the ADPr moiety to any nucleophilic amino acid acceptor with high confidence (*Hendriks et al., 2019*; *Larsen et al., 2018*). We used this MS platform to search for amino-acid-ADPr sites across the proteome in HEK 293 T cells under three experimental conditions: i. GFP-PARP-7 overexpressing cells, ii. MycX2-PARP-13.2 overexpressing cells, or iii. GFP-PARP-7 and MycX2-PARP-13.2 overexpressing cells. We identified a total of 1712 ADPr sites with a localization probability of >90% in all three conditions (*Supplementary file 1*). There is a low degree of variance between biological replicates, especially for the GFP-PARP-7 alone and the MycX2-PARP-13.2 alone conditions (*Figure 5—figure supplement 1*). As expected, there was a greater degree of overlap between the GFP-PARP-7 alone condition and the GFP-PARP-7/ MycX2-PARP-13.2 condition compared to the MycX2-PARP-13.2 condition (*Figure 5—figure supplement 2*). We identified amino acid-ADPr sites on all known ADPr acceptor residues, including Cys, Glu, Asp, His, Lys, Arg, Ser, Thr, and Tyr (*Supplementary file 1*).

Initially, we focused on analyzing the amino acid-ADPr sites in PARP-13. In the GFP-PARP-7 alone condition, endogenous PARP-13 (likely representing the two major isoforms PARP-13.1 and PARP-13.2) is MARylated exclusively on Cys residues (*Figure 5a*, *Figure 5—figure supplements 3–4*, *Supplementary file 1*). Intriguingly, all but one of the seven identified Cys-ADPr sites in endogenous PARP-13 are located in the four N-terminal zinc-finger (ZnF 1–4) domains; one Cys-ADPr (Cys721) is located in the inactive, catalytic domain of PARP-13.1 (*Figure 5b,c*, *Figure 5—figure supplement 3*, *Supplementary file 1*). Five out of the six ZnF Cys-ADPr sites are Zn-coordinating Cys residues (*Figure 5c*). Despite the relatively low conservation of PARP-13 across species, all of the endogenous PARP-13 Cys-ADPr sites in the ZnF domains are conserved between human, mouse, and rat (*Figure 5—figure supplement 5*).

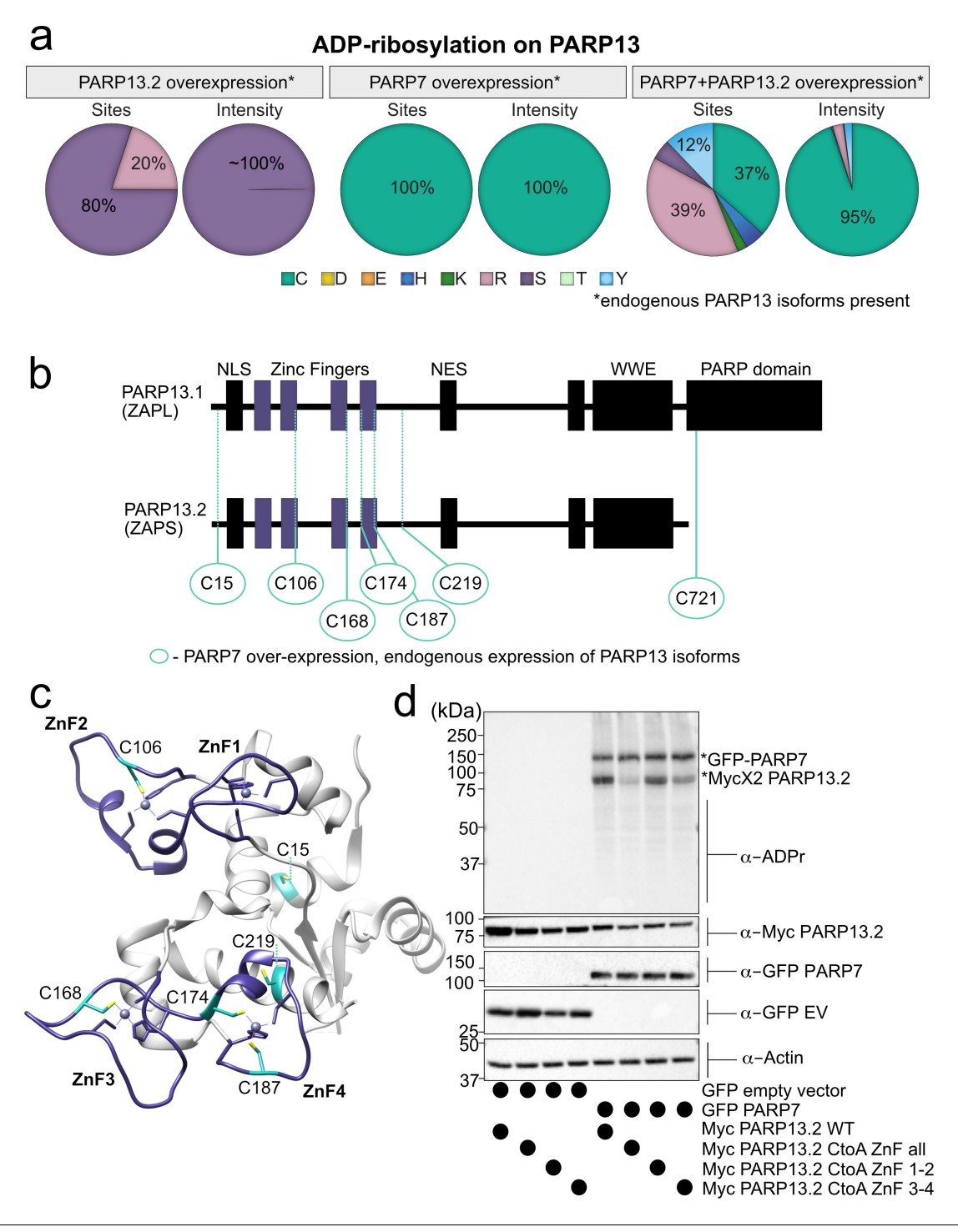

**Figure 5.** MS-based ADPr site identification and mutagenesis studies demonstrate that PARP-7-mediated MARylation of PARP-13 occurs predominately on Cys residues in the ZnF domains. (a) Pie-charts showing of the amino acid residue distribution of all ADP-ribosylation sites identified on PARP-13 using EThcD fragmentation. Experiments were performed under three conditions: MycX2-PARP-13.2 overexpressing cells, GFP-PARP-7 overexpressing cells, GFP-PARP-7 and MycX2-PARP-13.2 overexpressing cells. One letter code used for amino acids. (b) Domain architecture of PARP-13.1 and PARP-13.2 showing positions of Cys ADPr sites. (c) Crystal structure of the N-terminal ZnF domains of PARP-13highlighting Cys ADPr sites (cyan). ZnF domains are highlighted (purple). PDB: 6UEI. (d) Mutation of the Cys to Ala in the ZnF domains of PARP-13 decreased PARP-7-mediated PARP-13 MARylation. GFP-PARP-7 was co-expressed with either WT- or CtoA ZnF all, CtoA ZnF1-2, or

*Figure 5 continued on next page*

*Figure 5 continued*

CtoA ZnF3-4 in HEK 293 T cells. GFP-PARP-7 or GFP-empty vector control co-expressed in HEK293T cells with MycX2-PARP-13.2 WT and mutants. Proteins were resolved by SDS-PAGE and were detected by Western blot using antibodies against ADPr, GFP, Myc, and actin. Representative figure of data collected over three biological replicates.

The online version of this article includes the following figure supplement(s) for figure 5:

**Figure supplement 1.** EThcD sample correlation data.
**Figure supplement 2.** Scaled Venn diagram depicting distributions of ADP-ribosylation sites across samples.
**Figure supplement 3.** All ADPr sites on PARP-13 identified by EThcD fragmentation.
**Figure supplement 4.** Annotated EThcD MS/MS spectra.
**Figure supplement 5.** Alignment of PARP-13 human with rat and mouse with PARP-7 MARylation sites labeled.
**Figure supplement 6.** All ADPr sites identified by EThcD MS on PARP-7.
**Figure supplement 7.** Sequences of CtoA PARP-13.2 mutants.
**Figure supplement 8.** Mutation of Cys residues in the ZnF domain of PARP-13.2 do not disrupt PARP-7 binding.

When PARP-13.2 was co-expressed with PARP-7, we identified additional Cys-ADPr sites in the ZnF in PARP-13. In this condition, other amino acid-ADPr sites are identified, most prominently on Arg and Tyr; however, the intensities for Arg-ADPr and Tyr-ADPr are at least an order of magnitude lower than the intensities found for Cys-ADPr (*Figure 5a*, *Figure 5—figure supplement 3*). Cys MARylation in PARP-13.2 is dependent on the co-expression of PARP-7 since we only detect Ser-ADPr (three sites) and Arg-ADPr (one site) in the PARP-13.2 alone condition (*Figure 5a*, *Figure 5—figure supplement 3*). The intensity for Ser-ADPr is substantially higher than Arg-ADPr (*Figure 5a*, *Figure 5—figure supplement 3*).

In addition to identifying Cys-ADPr sites in PARP-13, we also found that GFP-PARP-7 is auto-MARylated predominately on Cys residues (*Figure 5—figure supplement 6*, *Supplementary file 1*). One Cys, Cys39, was identified in a previous MS method using standard HCD fragmentation (*Gomez et al., 2018*). Two Cys-ADPr sites (Cys543 and Cys552) reside in the catalytic domain of PARP-7. At much lower intensities, we found several Tyr-ADPr, His-ADPr, and Arg-ADPr sites in PARP-7 (*Figure 5—figure supplement 6*, *Supplementary file 1*). These lower intensity ADPr-sites could be sites of auto-MARylation or could be sites of trans-MARylation by other PARP family members. Regardless, it is clear that the major auto-MARylation sites in PARP-7 are Cys residues.

Using EThcD fragmentation, we confidently identified Cys residues as major PARP-7-mediated MARylation sites in PARP-13. Nevertheless, we sought to confirm the MS results using mutagenesis studies. We focused on the Cys residues in the ZnF domains since we found that they are the major sites of Cys MARylation in PARP-13 (*Figure 5*, *Figure 5—figure supplement 3*, *Supplementary file 1*). In HEK 293 T cells expressing both GFP-PARP-7 and MycX2-PARP-13.2, we found that mutation of all of the Cys residues to alanine (Ala) residues in the ZnF domains of PARP-13.2 (CtoA ZnF all) substantially reduces PARP-13.2 MARylation compared to WT PARP-13.2 (*Figure 5d*, *Figure 5—figure supplement 7*). Mutation of the Cys residues to Ala residues in either ZnF domains 1–2 (CtoA ZnF 1–2) or 3–4 (CtoA ZnF 3–4) modestly reduces PARP-13.2 MARylation (*Figure 5d*, *Figure 5—figure supplement 7*). The reduction in PARP-7-mediated MARylation of PARP-13.2 in the Cys mutants is not due to a loss in the interaction between PARP-13 and PARP-7 as shown by co-immunoprecipitation experiments (*Figure 5—figure supplement 8*). Together these data support that the primary sites of MARylation by PARP-7 on PARP-13.2 are the ZnF Cys residues.

## Proteome-wide analysis reveals Cys residues as major ADPr acceptors in PARP-7 targets

Having validated Cys as the major PARP-7-mediated MARylation site in PARP-13, we broadened our analysis across the proteome in HEK 293 T cells and focused on amino acid-ADPr sites that were either unique or enriched (at least 5-fold) in the GFP-PARP-7 alone condition compared to the MycX2-PARP-13.2 alone condition. We reasoned that these amino acid-ADPr sites are likely targets of PARP-7. Using the above criteria, we identified a total of 939 unique amino acid-ADPr sites on a total of 490 proteins (*Figure 6a*) as putative PARP-7 targets. Of these sites, the majority (471 Cys-ADPr sites on 374 proteins) were on Cys residues (*Figure 6a*). When we compared the PARP-7-targeted, Cys-ADPr modified proteins to the direct PARP-7 targets identified using our CG strategy,

we found 48 overlapping protein targets (*Figure 6b*). Among these 48 overlapping proteins, PARP-13 was a top target. We next asked how many Cys-ADPr sites there are per protein target. We found that the vast majority of proteins are MARylated on a single Cys (*Figure 6d* and *Supplementary file 1*). Intriguingly, PARP-13, with 11 sites identified, stands out as the only target with >4 Cys-ADPr sites (*Figure 6d*).

In some cases, enzymes that catalyze PTMs are directed to the target amino acid site by proximal consensus sequence motifs. Previous proteome-wide studies analyzing Ser-ADPr sites revealed a strong preference for a Lys residue preceding the Ser (−1 position relative to Ser-ADPr) that is ADP-ribosylated (*Leidecker et al., 2016*).We wondered if a similar preference might exist for Cys-ADPr. For this analysis, we used Icelogo, which can identify and visualize conserved patterns in proteins (*Colaert et al., 2009*). While we did not observe a strong preference for a particular amino acid at the −1 position, we did find a preference for a proline (Pro), followed by an Arg, in the +two position (*Figure 6c*). This analysis suggests that the proximal sequence surrounding the Cys-ADPr site is not a strong determinant for MARylation targeting by PARP-7.

## Discussion

We have successfully adapted our chemical genetic strategy for identifying the direct targets of PARP-7. This involved changing the position of the alkyne clickable handle from the N-6-position (5-Bn-6-a-NAD$^+$) to the 2-positon (5-Bn-2-e-NAD$^+$) of the adenosine ring in our modified NAD$^+$ analogs. 5-Bn-2-e-NAD$^+$ is a much better substrate for IG-PARP-7 compared to 5-Bn-6-a-NAD$^+$. Similarly, we found that 5-Bn-2-e-NAD$^+$ is a better substrate for floor position engineered PARP-12, I660G PARP-12 (IG-PARP-12) compared to 5-Bn-6-a-NAD$^+$ (*Figure 1—figure supplement 1b*). The 5-Bn-2-e-NAD$^+$—IG-PARP-12 pair will enable us, in future studies, to identify the direct targets of PARP12.

Combining our chemical genetics approach with a proximity labeling approach (BioID), we generated a list of PARP-7 MARylation targets and interactors with known roles in innate immune signaling. These results align with previous literature that demonstrate a critical role for PARP-7 as a negative regulator of interferon signaling during viral infection (*Yamada et al., 2016*; *Kozaki et al., 2017*; *Grunewald et al., 2020*). Particularly interesting is the enrichment of RNA binding and RNA regulatory proteins with known roles in viral regulation. Indeed, we found that the antiviral RNA-binding protein PARP-13 is a major MARylation target of PARP-7 in cells. In a co-submitted study, Kraus and colleagues also identified PARP-13 as PARP-7 target using a distinct, but complementary chemical genetic approach (*Palavalli Parsons et al., 2021*). Chemical sensitivity studies using HgCl$_2$ and MS-based site identification studies show that PARP-7 predominately MARylates PARP-13 on several Cys residues, most prominently in the ZnF domains of both PARP-13 isoforms, PARP-13.1 and PARP-13.2.

How might Cys MARylation regulate PARP-13 function? The most well-characterized function for PARP-13 is as an antiviral restriction factor. PARP-13 inhibits the replication of many types of RNA viruses (*Fehr et al., 2020*; *Daugherty et al., 2014*; *Todorova et al., 2015*; *Gao et al., 2002*; *Hayakawa et al., 2011*; *Karki et al., 2012*; *MacDonald et al., 2007*; *Lee et al., 2013*; *Mao et al., 2013*; *Müller et al., 2007*). Recent studies show that PARP-13.1 and PARP-13.2 have distinct roles in innate antiviral immune response: PARP-13.2 binds and degrades host mRNAs (e.g. *IFN-β*), thereby negatively regulating the interferon response, whereas PARP-13.1 targets viral RNA and is the major antiviral effector (*Schwerk et al., 2019*). The ZnF domains of PARP-13.1 and PARP-13.2 are required for binding to both host and viral mRNAs. Since the Cys residues in the ZnF domains of PARP-13 are important for Zn$^{2+}$ coordination, it is possible that MARylation of these Cys residues alters or disrupts RNA binding. The ZnF domains of PARP-13.2 are also involved in PPIs relevant to innate immune signaling. PARP-13.2 stimulates the interferon response in response to influenza A viral infection via direct activation of the cytosolic nucleic acid sensor RNA helicase RIG-I (*Hayakawa et al., 2011*). PARP-13.2 interacts with RIG-I and stimulates its oligomerization and ATPase activity, leading to interferon induction. This interaction is dependent on the ZnF domains of PARP-13.2, hence Cys MARylation of PARP-13.2 by PARP-7 could potentially disrupt the interaction between PARP-13.2 and RIG-I. Future studies will be focused on understanding how PARP-7-mediated Cys MARylation of PARP-13.1 and/or PARP-13.2 regulate their antiviral and immune regulatory roles.

Beyond PARP-7-mediated Cys MARylation of PARP-13, PARP-13 is also modified by other PARP family members at different amino acid acceptors. In a previous chemical genetic study, we found

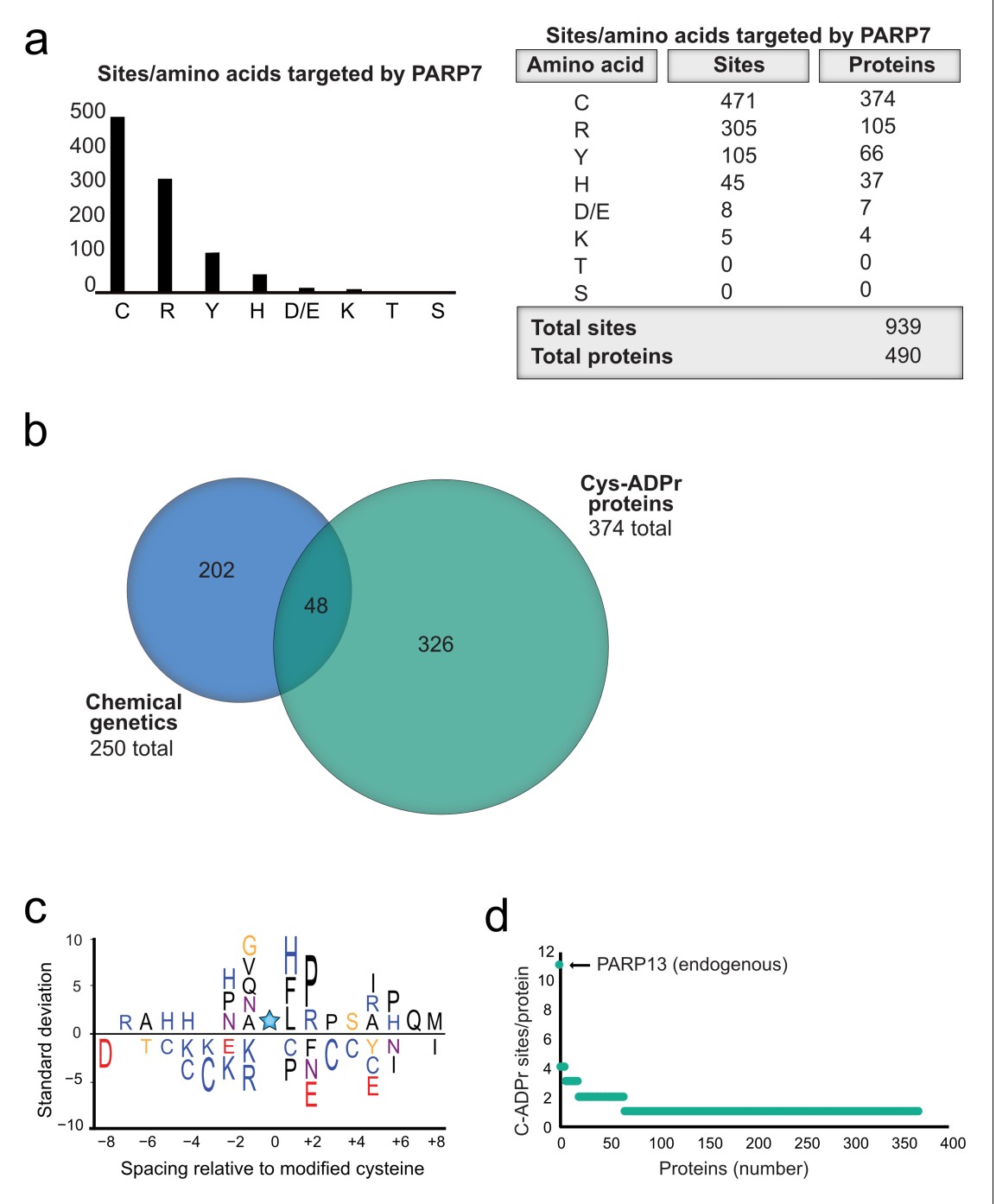

**Figure 6.** Proteome-wide analysis reveals Cys ADPr as a major MARylation site catalyzed by PARP-7. (a) Graph and corresponding table showing unique ADPr sites (and proteins) enriched in the PARP-7 overexpression condition (compared to MycX2-PARP-13.2 overexpression condition). (b) Venn diagram depicting the overlap of total proteins identified in our CG strategy compared to total Cys ADPr modified proteins enriched in the PARP-7 condition. (c) IceLogo representation of the sequence context surrounding Cys ADPr sites identified in GFP-PARP-7 overexpressing cells. Amino acids displayed above the line are enriched, and those displayed below are depleted, compared to the sequence context of all cysteine residues derived from the same proteins p=0.05. (d) Graph representing the number of endogenous Cys ADPr sites per protein in the PARP-7 overexpression sample. PARP-13, by far has the highest number of cysteine sites, with 11 sites identified.

that PARP14 MARylates PARP-13 on several Glu/Asp residues in the C-terminus of the protein (*Carter-O'Connell et al., 2018*). Another study showed that PARP-13.1 and PARP-13.2 can be poly-ADP-ribosylated (PARylated) in response to cell stress, although the PARP that mediates PARylation

of PARP-13.1/2 was not identified (*Leung et al., 2011*). Interestingly, in this study, we find that in the absence of PARP-7, PARP-13.2 is ADP-ribosylated on several serines. Serines are known sites of PARylation by PARP1 and PARP2 (*Larsen et al., 2018*; *Palazzo et al., 2018*); therefore, PARylation of PARP-13 could be catalyzed by PARP1/2. Trans-M/PARylation of PARP-13 by other PARP family members suggests that PARP-13 is an important integrator of active PARP signaling. An intriguing idea is that M/PARylation at distinct amino acid acceptors elicits different downstream effects.

Proteome-wide ADPr site profiling suggests that many targets of PARP-7 are MARylated predominately on Cys residues. Cys MARylation was originally described for the bacterial ADP-ribosyltransferase Pertussis toxin (PTX) (*Mangmool and Kurose, 2011*). PTX MARylates a Cys in the carboxyl terminus of the alpha submit of the G proteins, $G_i$ and $G_o$ (*Mangmool and Kurose, 2011*). Studies dating back 30 years ago described endogenous Cys MARylation, similar to PTX-dependent MARylation (*Jacobson et al., 1990*; *Tanuma, 1988*) but the enzyme(s) that catalyzes Cys MARylation were not identified. In this study, we provide strong evidence that PARP-7 catalyzes Cys MARylation. Our results are consistent with previous studies showing that PARP-7 auto-MARylates predominately on a Cys residue (Cys39) (*Gomez et al., 2018*). How PARP-7 preferentially targets Cys versus other potential amino acid acceptors is currently unknown. Recent studies on the amino acid selectivity of PARP1/2-mediated PARylation provides some clues: for example, a recent crystal structure of the histone PARylation factor (HPF1) (*Gibbs-Seymour et al., 2016*)—PARP2 complex shows that HPF1 interacts directly with the $NAD^+$-binding site of PARP2 (*Suskiewicz et al., 2020*), and demonstrates that HPF1 can act as an auxiliary factor to influence amino acid target preference. In the absence of HPF1, PARP1 predominately PARylates on Glu/Asp residues; however, in the presence of HPF1, PARP1 predominately PARylates on Ser residues (*Palazzo et al., 2018*; *Eisemann et al., 2019*). Perhaps this is similarly the case for PARP-7, and future studies may focus on identifying potential auxiliary factors for PARP-7.

Curiously, inhibition of PARP-7 catalytic activity by Phthal01 increased endogenous PARP-7 protein levels. This is consistent with previous studies showing that a catalytically inactive PARP-7 mutant is expressed at higher levels than WT-PARP-7 (*Gomez et al., 2018*). Although endogenous PARP-13.1 levels were not impacted by Phthal01, it is possible that PARP-7 regulates the stability of PARP-13.2 in a context-specific manner, for example, during viral infection. Future studies will focus on determining the mechanism by which inhibition of PARP-7 catalytic activity regulates PARP-7 protein levels, and if PARP-13.2 protein levels (or other PARP-7 targets) are regulated by PARP-7 trans-MARylation.

Until recently, PARP drug discovery efforts over the past 20 years have focused exclusively on PARP1/2 and its role in the DNA damage response. The role of PARP-7 in the innate immune response has stimulated interests in PARP-7 as an immunomodulatory agent for cancer treatment. Currently, there is the first PARP-7 inhibitor (RBN-2397) in Phase I clinical trials for solid tumor cancers (NCT04053673). This is the first clinical trial for an inhibitor of a MARylating PARP and represents and exciting new direction in the PARP field. The PARP-7 targets and ADPr sites identified in this study as well as the Parsons et al. study will guide future studies examining the mechanism by which PARP-7 regulates antitumor immunity.

# Materials and methods

**Key resources table**

| Reagent type (species) or resource | Designation | Source or reference | Identifiers | Additional information |
|---|---|---|---|---|
| Antibody | Anti-ADP-ribose, E6F6A (Rabbit monoclonal) | Cell signaling technology | 83732: RRID:AB_2749858 | WB(1:1000) |
| Antibody | Anti-GFP, PABG1 (Rabbit polyclonal) | Chromotek | PABG1-100; RRID:AB_2749857 | WB(1:1000) |
| Antibody | Anti-HA.11, 16B12 (Mouse monoclonal) | BioLegend | 901502; RRID:AB_2565007 | WB(1: 2000) |
| Antibody | Anti-Myc, 9B11 (Mouse monoclonal) | Cell signaling technology | 2276S; RRID:AB_331783 | WB(1:1000) |

*Continued on next page*

*Continued*

| Reagent type (species) or resource | Designation | Source or reference | Identifiers | Additional information |
|---|---|---|---|---|
| Antibody | Anti-β-Actin, C-4 (Mouse monoclonal) | Santa Cruz Biotechnology | sc-47778; RRID:AB_2714189 | WB(1:1000) |
| Antibody | Anti-ZC3HAV1/PARP-13 (Rabbit polyclonal) | Proteintech | 16820–1-AP; RRID:AB_2728733 | WB(1:1000) |
| Antibody | Goat anti-rabbit IgG, HRP linked | Jackson Immuno Research Lab | 20402–1 mL | WB(1:10000) |
| Antibody | Anti rabbit IgG, HRP linked | Cell signaling technology | 7074S | WB(1:1000) |
| Antibody | Goat anti-mouse IgG (H+L), HRP linked | Invitrogen | 62–6520 | WB(1:5000) |
| Antibody | Anti-ZC3HAV1/PARP-13 (Rabbit polyclonal) | Invitrogen | PA5-106389; RRID:AB_2854063 | WB(1:1000) |
| Antibody | Anti-ZC3HAV1, N3C2 (Mouse monoclonal) | GeneTex | GTX120134; AB_10721153 | WB(1:2,000) |
| Antibody | Anti- TIPARP/PARP-7 (mouse monoclonal) | This paper | | WB(1:1000) |
| Peptide, recombinant protein | Streptavidin HRP (Peroxidase Streptavidin 1.0 mg) | Jackson Immuno Research Lab | NC9705430 | WB(1:2500) |
| Other | GFP-Trap Magnetic Agarose | Chromotek | gtma-20 | |
| Other | Myc-trap magnetic agarose | Chromotek | ytma-10 | |
| Other | Pierce NeutrAvidin Agarose | Thermo Scientific | 29201 | |
| Other | Cytiva (Formerly GE Healthcare Life Sciences) Glutathione Sepharose 4B Media | Fisher Scientific | 45-000-139 | |
| Chemical compound, drug | Biotin-Peg3-Azide | Click Chemistry Tools | AZ104-5 | |
| Chemical compound, drug | Tris (3-hydroxypropyltriazolylmethyl) amine (THPTA) | Sigma-Aldrich | 762342–100 MG | |
| Chemical compound, drug | cOmplete EDTA-free Protease Inhibitor Cocktail | Sigma-Aldrich | 11873580001 | |
| Chemical compound, drug | Hydroxylamine solution | Sigma-Aldrich | 438227–50 ML | |
| Chemical compound, drug | Fetal bovine serum | Sigma-Aldrich | TMS-013-B | |
| Chemical compound, drug | Gibco DMEM- high glucose | Fisher Scientific | 11965092 | |
| Chemical compound, drug | Gibco GlutaMAX supplement | Fisher Scientific | 35050061 | |
| Chemical compound, drug | 5bn-6a-NAD+ | *JACS*, **136**, 5201 (2014) | | |
| Chemical compound, drug | 5bn-2e-NAD+ | This paper | | See Methods Supplement |
| Chemical compound, drug | Phthal01 | This paper | | See Methods Supplement |
| Recombinant DNA reagent | Myc-BirA*-PARP-7 | This paper | | Cloned form Myc-BirA*-PARP-14 described in *ACS Chem. Biol.*, **13**, 2841 (2018) |
| Recombinant DNA reagent | GFP-WT-PARP-7 | This paper | | WT-PARP-7 gene (gBlock gene fragment) was cloned into pEGFP-C1 |

*Continued on next page*

*Continued*

| Reagent type (species) or resource | Designation | Source or reference | Identifiers | Additional information |
|---|---|---|---|---|
| Recombinant DNA reagent | GFP-IG-PARP-7 | This paper | | IG-PARP-7 gene (gBlock gene fragment) was cloned into pEGFP-C1 |
| Recombinant DNA reagent | HA-PARP-13.1 | *ACS Chem. Biol.*, **13**, 2841 (2018) | | |
| Recombinant DNA reagent | HA-PARP-13.2 | This paper | | WT-HA-PARP-13.2 gene (gBlock gene fragment) was cloned into CMV vector using standard restriction enzyme methods |
| Recombinant DNA reagent | MycX2-PARP-13.2 | This paper | | WT-MycX2-PARP-13.2 gene (gBlock gene fragment) was cloned into CMV vector using standard restriction enzyme methods |
| Recombinant DNA reagent | MycX2-PARP-13.2 CtoA ZnF all mutant | This paper | | Mutant MycX2-PARP-13.2 gene (gBlock gene fragment) was cloned into CMV vector using standard restriction enzyme methods |
| Recombinant DNA reagent | MycX2-PARP-13.2 CtoA ZnF 1–2 mutant | This paper | | Mutant MycX2-PARP-13.2 gene (gBlock gene fragment) was cloned into CMV vector using standard restriction enzyme methods |
| Recombinant DNA reagent | MycX2-PARP-13.2 CtoA ZnF 3–4 mutant | This paper | | mutant MycX2-PARP-13.2 gene (gBlock gene fragment) was cloned into CMV vector using standard restriction enzyme methods |
| Recombinant DNA reagent | HA-MacroD2 | This paper | | HA-MacroD2 gene (gBlock gene fragment) was cloned into CMV vector using standard restriction enzyme methods |
| Recombinant DNA reagent | GFP-PARP-10 | *Cell Reports*, **25**, 4770 (2015) | | |
| Recombinant DNA reagent | mCherry | gift plasmid from RY Tsien | | |
| Recombinant DNA reagent | GST-PARP-7 | Gomez et al., *Biochemical Journal*, **475**, 3827 (2018) | | |
| Recombinant DNA reagent | His6-SUMO-PARP-13.2 | gift plasmid from I. Carter-O'Connell | | |

## Cell culture

HEK 293T and A549 (ATCC CCL-185) cells were grown in DMEM (Gibco) supplemented with 10% fetal bovine serum (FBS, Millipore Sigma), and glutamax (Gibco) at 37 °C and 5% CO2. Mouse embryonic fibroblasts (MEFs) derived from WT or KO PARP-7 mice were cultured as previously described (*Yamada et al., 2016*). HEK 293T and A549 cells were authenticated using STR and were found to be mycoplasma free. Transient transfections of HEK 293 T cells with 15 μg of expression vectors per 10 cm dish (70% confluency) were performed using the CalPhos system (Clontech) according to manufacturer's instructions. Cells were lysed in HEPES buffer supplemented with 1% Triton X-100, 100 μM fresh TCEP-HCl (Thermo Scientific), cOmplete EDTA-free protease inhibitor (Roche), phosphatase inhibitor cocktail 2 (Sigma-Aldrich), and phosphatase inhibitor cocktail 2 (Sigma-Aldrich). Cell debris was cleared by centrifugation at 10,000 X g for 5 min at 4C.

## PARP-7 MARylation target labeling and neutravidin enrichment for LC-MS/MS analysis

Lysates from HEK 293 T cells expressing WT- or IG-PARP-7 (1.169 mg) were diluted to a volume of 336 µl in lysis buffer (25 mM HEPES pH 7.4, 50 mM NaCl, 5 mM MgCl$_2$, 1% NP-40, 1X cOmplete EDTA-free protease inhibitor (Roche), 0.5 mM TCEP). The lysates were then split into six tubes of 48 µl. To each tube, 12 µl of 5X concentration (0.5 mM) 5-Bn-2e-NAD$^+$ was added for a final concentration of 100 µM in each tube. The samples were then incubated for 2 hr shaking at 30 ˚C. Following incubation, the samples were precipitated with 1 ml 4:1 cold MeOH: CHCl3 for 1 hr at −20˚C to remove excess 5-Bn-2e-NAD$^+$. This step was essential before proceeding to the click reaction. The protein was then pelleted at 6000 g at 4˚C for 30 min. Methanol supernatant was completely removed and the pellet was allowed to dry for approximately 15 min. The pellet was resuspended in 30 µl of 2% SDS and 30 µl of PBS was added. To this 30 µl of 3X concentration click buffer was added (15.8 mM THPTA in PBS, 3.15 mM CuSO4 made fresh in 1X PBS, 0.75 mM Biotin-peg3-azide (Click Chemistry Tools) in DMSO, 15.8 mM sodium ascorbate in PBS, diluted to respective concentrations with 1X PBS). Click conjugation to biotin−peg3-azide was performed for 1 hr at RT. Following click conjugation, WT-PARP-7 samples were pooled and IG-PARP-7 samples were pooled and precipitated overnight with 8 mL cold MeOH at 20˚C. The next day, samples were pelleted at 6000 g 4˚C. Importantly, the samples were pelleted, sonicated for 5 s, resuspended in cold methanol and pelleted again. This process was repeated three times with cold methanol and once with cold acetone. This step was essential to remove excess biotin-Peg3-azide before enrichment. The protein was then re-dissolved in 2% SDS and subjected to enrichment using NeutrAvidin agarose (Pierce) and proteolysis as previously described (*Carter-O'Connell et al., 2014*; *Carter-O'Connell et al., 2016*; *Carter-O'Connell and Cohen, 2015*; *Carter-O'Connell et al., 2018*).

## CG and BioID proteomics methods

All MS methods and data analysis were followed as previously reported (*Carter-O'Connell et al., 2014*; *Carter-O'Connell et al., 2016*; *Carter-O'Connell and Cohen, 2015*; *Carter-O'Connell et al., 2018*). For both the CG method or BioID method, a protein was considered a 'valid' target (CG) or interactor (BioID) if the unique peptide count of the identified protein was at least twice as abundant in the IG-PARP-7 or Myc-BirA*-PARP-7 sample in comparison to the control sample (GFP-PARP-7).

## In cell MARylation of MycX2-PARP-13.2 by GFP-PARP-7

HEK293T cells were seeded and grown overnight to ~60% confluency on a six-well plate. The next morning the cells were transfected using CalPhos Mammalian Transfection kit. A total of 1.5 µg of each plasmid (for co-transfection) were used in each well of the six-well plate and transfection protocol according to manufacturer was followed. After 4–6 hr transfection, the media was swapped with fresh warm media. The cells were allowed to grow overnight and typically reached ~90% confluency. The following morning, media was aspirated and cells were washed with 2 ml cold PBS/ well. All PBS were aspirated and the cell plate was frozen in the −80 ˚C until ready to lyse. Cell plates from −80 ˚C were taken out onto ice and 75 µl of LysB (50 mM HEPES pH7.4, 150 mM NaCl, 1 mM MgCl$_2$, 1% tritonX-100, 1X protease inhibitor, 1X phosphatase inhibitors, 1 µM veliparib, 100 µM TCEP) added to the frozen wells of the plate. The plates were allowed to thaw for about 10 min on ice and the cells were removed from the plate by pipetting up and down in the wells to collect cell lysates. Lysates were transferred to Eppendorf tubes and centrifuged 10,000 g for 5 min at 4˚C. The supernatant was transferred to a fresh tube and the protein concentration was determined by the Bradford assay (Biorad). The lysates were brought to 200 µg/60 µl and then 20 µl of 4X sample buffer was added for a final concentration of 2.5 µg/ul. The samples were boiled and 15 µl (37.5 µg) of total proteins were separated by 10% SDS-PAGE. Proteins were then transferred to nitrocellulose, blocked with milk and probed for ADPr (CST) 1:1000, GFP (Chromotek, 1:1000), actin (Santa Cruz Biotechnology, 1:1000), and Myc (CST, 1:1000).

## In vitro MARylation with bacterially expressed PARPs

GST-PARP-7, His$_6$-SUMO-PARP-13.2, and His$_6$-SUMO-PARP-10 were expressed (*E. coli*) and purified as previously described (*Kirby et al., 2018*). MARylation reaction conditions: 10 nM PARP-10 or PARP-7, 30 nM PARP-13.2, and 100 µM NAD$^+$ in HEPES reaction buffer (HRB: 50 mM HEPES, 100 mM NaCl,

10 mM MgCl$_2$, and 0.5 mM TCEP). The MARylation reactions were allowed to proceed at 30°C and stopped at various time intervals by the addition of sample buffer. The samples were boiled and proteins were separated by 10% SDS-PAGE. Proteins were transferred to nitrocellulose, blocked with milk and probed for ADPr (CST, 1:1000). Coomassie stain was used as a protein loading control.

## GST pulldown of recombinant PARP-13.2 by recombinant PARP-7

His$_6$-SUMO-PARP-13.2 (100 nM) and GST-PARP-7 (100 nM) were incubated in HRB for 1 hr at 4°C. The above mixture was then incubated with either vehicle control, NAD$^+$ (100 µM), Phthal01 (1 µM), or NAD$^+$ together with Phthal01 (100 µl/sample). After 2 hr at 30°C, protein complexes were enriched using GSH-Sepharose (Cytiva) (50 µl 50% slurry, rotation, 1 hr at 4°C). The beads were washed with HRB containing 1 mM EDTA (3x). The proteins were eluted with 1.5x Sample buffer (80 µl) and boiled for 10 min at 95°C. Proteins were resolved by 7.5% SDS-PAGE, transferred to nitrocellulose, blocked with milk and probed for ADPr, GST (Proteintech, 1:2000), or His$_6$ (Fisher Scientific, 1:1000).

## Chemical treatment with HgCl$_2$ and NH$_2$OH

HEK293T cells co-expressing either GFP-PARP-7 and mycX2-PARP-13.2 or GFP-PARP-10 and mycX2-PARP-13.2 were lysed and protein was quantified by Bradford assay and brought to a concentration of ~3 mg/ml. Lysates were then treated with 1% SDS to prior to lysis to prevent to inactivate enzymes. Lysates were then treated either with H$_2$O (control), 2 mM HgCl$_2$ in H$_2$O (1.5 hr) in ddH2O, 0.4 M neutral NH$_2$OH in H$_2$O (15 min). Following treatment, proteins were precipitated by adding cold methanol and incubating samples at −20°C. Following precipitation with cold methanol, proteins were centrifuged at 6000 g. The methanol was removed and protein pellets were allowed to dry for ~10 min at RT. The protein was resuspended in 1.5X sample buffer containing βME and boiled for 10 min at 95°C. Proteins were separated by 10% SDS-PAGE, transferred to nitrocellulose, and western blot analysis was performed to detect ADPr, GFP-PARPs, mycX2-PARP-13.2, and actin.

## PARP-7 antibody validation and effects of Phthal01 on endogenous PARP-7 levels in cell culture

A549 cells or WT or KO MEFs PARP-7 were plated at a density of $2.0 \times 10^5$ cell per ml in six-well plates. The following day, the cells were treated for 4 hr with DMSO, 10 nM TCDD, 1 µM Phthal01, or co-treated with TCDD and Phthal01. Cell pellets were collected and lysed in RIPA buffer (20 mM Tris-HCl (pH 7.5), 150 mM NaCl, 1 mM EDTA, 1 mM EGTA, 1% NP-40, 1% sodium deoxycholate) supplemented with 1X protease inhibitor cocktail (Roche). Samples were sonicated at a low intensity for two cycles of 30 s on/off two times using a Bioruptor and rotated for 30 min at 4°C. After centrifugation, the protein concentration was determined by BCA assay (Bio-Rad). 40 µg of total protein was separated by SDS-PAGE and transferred to a PVDF membrane. Membranes were incubated with in house generated anti-PARP-7 (1:1000) or anti-α-actin in 5% milk overnight. Membranes were stripped, blocked, and re-blotted with anti-PARP-13 (Invitrogen PA5-106389; 1:1000) or with anti-PARP-7 (Abcam 84664 lot# GR3304056-5; 1:1000).

## Sample preparation for EThcD MS analysis

Samples were overall prepared as previously described (*Hendriks et al., 2019*; *Larsen et al., 2018*). Cell pellets were lysed in guanidinium (6 M guanidine-HCl, 50 mM TRIS, pH 8.5) by alternating vigorous vortexing and vigorous shaking of the samples, after which lysates were snap frozen in liquid nitrogen. Cell lysates were thawed at room temperature, reduced and alkylated by incubation with 5 mM TCEP and 5 mM chloroacetamide (CAA) for 30 min, followed by sonication for 15 s at an amplitude of 90%. Protein concentration was measured using Bradford assay (Bio-Rad). Proteins were digested for 3 hr at room temperature using Lysyl Endopeptidase (Lys-C, 1:100 w/w; Wako Chemicals). After initial digestion, samples were diluted with three volumes of 50 mM ammonium bicarbonate (ABC), after which they were digested using modified sequencing grade Trypsin (1:100 w/w; Sigma Aldrich) overnight at room temperature.

Subsequently, protease digestion was terminated by addition of trifluoroacetic acid (TFA) to a final concentration of 0.5% (v/v), and cleared from precipitates by centrifugation in a swing-out centrifuge at 4°C, for 45 min at 4250 g. Peptides were purified by reversed-phase C18 cartridges (Sep-Pak, Waters), which were pre-activated with 5 mL ACN, and equilibrated 2X with 5 mL of 0.1% TFA.

After sample loading, cartridges were washed with 3X with 10 mL of 0.1% TFA, after which peptides were eluted using 4 ml of 30% ACN in 0.1% TFA. Eluted peptides were frozen at −80°C overnight, after which they were dried to completion by lyophilization.

Lyophilized peptides were dissolved in AP buffer (50 mM TRIS pH 8.0, 50 mM NaCl, 1 mM MgCl2, and 250 µM DTT), and cleared by centrifugation in a swing-out centrifuge at room temperature, for 30 min at 4250 g. Poly-ADP-ribosylation was reduced to mono-ADP-ribosylation by incubation of samples with recombinant PARG (a kind gift from Prof. Michael O. Hottiger) at a concentration of 1:10,000 (w/w), at room temperature, overnight and with gentle sample agitation. After overnight incubation with PARG, samples were cleared from mild precipitation by centrifugation in a swing-out centrifuge at 4°C, for 60 min at 4250 g. Next, in-house prepared sepharose beads with GST-tagged Af1521 were added to the samples, in a ratio of 50 µl dry beads per 5 mg sample. Samples were incubated in a head-over-tail mixer, at 4°C for 3 hr. Afterwards, beads were washed twice in ice-cold AP Buffer, twice in ice-cold PBS, and twice with ice-cold MQ water. After each second washing step, tubes were changed in order to minimize non-specific carryover of contaminants. After washing, ADP-ribosylated peptides were eluted off the beads using two bead volumes of ice-cold elution buffer (0.15% TFA). The elution was performed by gentle addition of the elution buffer, gentle mixing of the beads with the buffer every 5 min, and otherwise allowing the beads to stand undisturbed on ice for 20 min. Beads were gently pelleted, and the elutions were transferred to 0.45 µm column filters (Ultrafree-MC, Millipore). Elution of the beads was repeated once, and the two elutions were combined on the 0.45 µm column filters. ADP-ribosylated peptides were then transferred through the filters by centrifugation for 1 min at 12,000 g. A total of 100 kDa cut-off filters (Vivacon 500, Sartorius) were pre-washed by surface-washing the filters with 300 µl of MQ water once, spinning 2 × 400 µl of MQ water through the filters, surface-washing the filters with 300 µl of 0.15% TFA once, spinning 400 µl of 0.15% TFA through the filters, replacing the collection tubes, and spinning 400 µl of 0.15% TFA through the filters once more. Next, ADP-ribosylated peptides were transferred to the pre-washed 100 kDa cut-off filters, and centrifuged for 10 min at 8000 g.

Samples were basified by addition of ammonium hydroxide to a final concentration of 20 mM. All StageTips were prepared essentially as described previously (*Rappsilber et al., 2003*) but were assembled using four layers of C18 disc material (punch-outs from 47mm C18 3M extraction discs, Empore). StageTips were activated using 100 µL methanol, and re-activated using 100 µl of 80% ACN in 50 mM ammonium hydroxide. StageTips were equilibrated using 2 × 100 µl of 20 mM ammonium hydroxide, after which samples were loaded. The flow-through was collected at this stage (F0). Subsequently, StageTips were washed twice with 100 µl of 20 mM ammonium hydroxide, of which the first wash was collected (F0). Samples were eluted of the StageTips using 80 µl of 30% of ACN in 20 mM ammonium hydroxide (F1). Flow-through from loading of the samples were acidified to a final concentration of 1% TFA (v/v), and then processed as above except using 0.1% formic acid instead of ammonium hydroxide. All samples were vacuum dried to completion in a SpeedVac at 60°C. Dried purified ADP-ribosylated peptides were dissolved by addition of 10 µl 0.1% formic acid, and stored at −20°C until MS analysis.

## EThcD MS data acquisition

All MS samples were measured using a Fusion Lumos Orbitrap mass spectrometer (Thermo). Samples were analyzed on 15 cm long analytical column, packed in-house using ReproSil-Pur 120 C18-AQ 1.9 µm beads (Dr. Maisch), with an internal diameter of 75 µm. On-line reversed-phase liquid chromatography to separate peptides was performed using an EASY-nLC 1200 system (Thermo). The analytical column was heated to 40°C using a column oven, and peptides were eluted from the column using a gradient of Buffer A (0.1% formic acid) and Buffer B (80% can in 0.1% formic acid). For the main samples (F1), the gradient ranged from 3% buffer B to 40% buffer B over 62 min, followed by a washing block of 18 min. For the flow-through samples (F0), the gradient ranged from 5% buffer B to 30% buffer B over 35 min, followed by a washing block of 15 min. Electrospray ionization (ESI) was achieved using a Nanospray Flex Ion Source (Thermo). Spray voltage was set to 2 kV, capillary temperature to 275°C, and RF level to 30%. For F1 samples, full scans were performed at a resolution of 60,000, with a scan range of 300–1750 m/z, a maximum injection time of 60 ms, and an automatic gain control (AGC) target of 600,000 charges. Precursors were isolated with a width of 1.3 m/z, with an AGC target of 200,000 charges, and precursor fragmentation was accomplished using electron transfer disassociation with supplemental higher collisional disassociation (EThcD) with a

supplemental activation energy of 20. Precursors with charge state 3–5 were isolated for MS/MS analysis, and prioritized from charge 3 (highest) to charge 5 (lowest), using the decision tree algorithm. Selected precursors were excluded from repeated sequencing by setting a dynamic exclusion of 60 s. MS/MS spectra were measured in the Orbitrap, with a loop count setting of 5, a maximum precursor injection time of 120 ms, and a scan resolution of 60,000. The F0 samples were measured as described above with the following exceptions. Full scans were performed at a resolution of 120,000, with a maximum injection time of 250 ms, and MS/MS spectra were measured at a resolution of 60,000, with a maximum precursor injection time of 500 ms.

### EThCD data analysis

All raw data analysis was performed using MaxQuant software (version 1.5.3.30) supported by the Andromeda search engine (*Rappsilber et al., 2003*; *Cox and Mann, 2008*) Default MaxQuant settings were used, with the following exceptions. Methionine oxidation, N-terminal acetylation, cysteine carbamidomethylation, and ADP-ribosylation on C, D, E, H, K, R, S, T, and Y, were included as variable modifications. Up to six missed cleavages were allowed, and a maximum allowance of 4 variable modifications per peptide was used. Second peptide search was enabled (default), and matching between runs was enabled with a match time window of 0.7 min and an alignment time window of 20 min. Mass tolerance for precursors was set to 20 ppm in the first MS/MS search and 4.5 ppm in the main MS/MS search after mass recalibration. For fragment ion masses, a tolerance of 20 ppm was used. Modified peptides were filtered to have an Andromeda score of >40 (default), and a delta score of >20. Data was automatically filtered by posterior error probability to achieve a false discovery rate of <1% (default), at the peptide-spectrum match, the protein assignment, and the site-specific levels.

In addition to the FDR control applied by MaxQuant, the data was manually filtered using the Perseus software (*Tyanova et al., 2016*) in order to ensure proper identification and localization of ADP-ribosylation. As default MaxQuant intensity assignments to modification sites also include non-localized or poorly localized evidences, intensities were manually mapped back to the sites table based on localized PSMs only (>0.90 best-case,>0.75 for further evidences). The iceLogo web application was used for sequence motif analysis (*Colaert et al., 2009*), and for sequence comparisons, background sequences were extracted from the same proteins and flanking the same amino acid residue type using an in-house Python script.

### Chemical synthesis

For the synthesis of 5-Bn-2-e-NAD$^+$ and Phthal01 see *Supplementary files*. Synthesis of 5-Bn-6-a-NAD$^+$ was described previously (*Carter-O'Connell et al., 2014*).

## Acknowledgements

We thank John Klimek and Philip Wilmarth for assistance with the CG and BioID MS analysis at the OHSU proteomics core. Mass spectrometry analysis performed by the OHSU Proteomics Shared Resource was partially supported by NIH grants P30EY010572, P30CA069533, and S10OD012246. We thank current and past members of the Cohen lab for many fruitful discussions regarding the experimental design, data analysis, and the preparation of the manuscript. We thank Jonas Damgaard Elsborg for writing the python script used for generation of background sequences for motif analyses, and Ivo Alexander Hendriks for fruitful discussions and help with data analysis of EThcD MS data. We thank Prof. W Lee Kraus for sharing their PARP-7 chemical genetic studies prior to publication. We thank Prof. Michael O Hottiger for the PARG plasmid. We thank Prof. Ian Carter-O'Connell for the SUMO-PARP-13.2 plasmid. This work was funded by the NIH (NIH 2R01NS088629) and the Pew Foundation to MSC.

## Additional information

### Funding

| Funder | Grant reference number | Author |
| --- | --- | --- |
| National Institute of Neurological Disorders and Stroke | NIH 2R01NS088629 | Michael S Cohen |

| Pew Charitable Trusts | | Michael S Cohen |
| NIH | 2R01NS088629 | Michael S Cohen |

The funders had no role in study design, data collection and interpretation, or the decision to submit the work for publication.

## Author contributions

Kelsie M Rodriguez, Data curation, Formal analysis, Investigation, Methodology, Writing - original draft; Sara C Buch-Larsen, Data curation, Formal analysis, Writing - review and editing; Ilsa T Kirby, Data curation; Ivan Rodriguez Siordia, Marit Rasmussen, Data curation, Methodology; David Hutin, Resources, Data curation, Methodology; Denis M Grant, Resources, Formal analysis, Methodology; Larry L David, Formal analysis; Jason Matthews, Resources, Data curation, Methodology, Writing - review and editing; Michael L Nielsen, Formal analysis, Methodology, Writing - review and editing; Michael S Cohen, Conceptualization, Formal analysis, Funding acquisition, Writing - original draft, Writing - review and editing

## Author ORCIDs

Kelsie M Rodriguez (iD) https://orcid.org/0000-0002-0821-6717
Sara C Buch-Larsen (iD) http://orcid.org/0000-0001-6250-5467
Michael S Cohen (iD) https://orcid.org/0000-0002-7636-4156

## Decision letter and Author response

Decision letter https://doi.org/10.7554/eLife.60480.sa1
Author response https://doi.org/10.7554/eLife.60480.sa2

# Additional files

## Supplementary files

• Supplementary file 1. See Excel file. Tab 1: PARP-7 MARylation targets identified from HEK 293 T cell lysates using a chemical genetic approach. Tab 2: PARP-7 interactors identified from HEK 293 T cells using proximity labeling (BioID approach) Tab 3: reviGO analysis, related to *Figure 1a*. Tab 4: Metascape GO analysis, related to *Figure 1b*. Tab 5: ADP-ribosylation targets and sites identified from HEK 293 T cells expressing PARP-13.2 alone, PARP-7 alone, or PARP-13.2 and PARP-7. Tab 6: Number of cysteine sites per protein target, related to *Figure 4d*.

• Transparent reporting form

## Data availability

All RAW proteomics files have been uploaded to PRIDE. This is related to all Supplementary Tables.

The following dataset was generated:

| Author(s) | Year | Dataset title | Dataset URL | Database and Identifier |
|---|---|---|---|---|
| Rodriguez KM, Buch-Larsen SC, Kirby IT, Siordia IR, Hutin D, Rasmussen M, Grant DM, David LL, Matthews J, Nielsen ML, Cohen MS | 2021 | Chemical genetics and proteome-wide site mapping reveal cysteine MARylation by PARP-7 on immune-relevant protein targets | https://www.ebi.ac.uk/pride/archive/projects/PXD020323 | PRIDE, PXD020323 |

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

## Appendix 1

### General Chemistry methods section

#### General

[1]H NMR were recorded on a Bruker DPX spectrometer at 400 MHz. Chemical shifts are reported as parts per million (ppm) downfield from an internal tetramethylsilane standard or solvent references. Dichloromethane (DCM), tetrahydrofuran (THF), and Dimethylformamide (DMF) were dried using a solvent purification system manufactured by Glass Contour, Inc (Laguna Beach, CA). Additional drying of solvents, where indicated in methods, occurred using 3Å or 4Å activated sieves. All other solvents were of ACS chemical grade (Fisher Scientific) and used without further purification unless otherwise indicated. Commercially available chemical compounds were purchased from Combi-Blocks (San Diego, CA), and TCI America (Portland, OR) and were >95% pure and used without further purification. All other reagents were of ACS chemical grade (Fisher Scientific) and used as received. 5-Bn-6-a-NAD[+] was synthesized as previously described (*Carter-O'Connell et al., 2016*). HPLC Methods: Gradient 10 mM tributylamine/30 mM acetic acid pH 4.4 to 100% methanol over 16 mins using c18 column.

**Appendix 1—scheme 1.** Synthesis of 5-bn-2-ethynyl NAD[+]. (**a**) p-TsOH, acetone, sodium carbonate; (**b**) (PPh$_3$) PdCl$_2$, CuI, TEA, TMS-acetylene; (**c**) 7N NH$_3$/MeOH; (**d**) POCl$_3$,TEA, THF, 0 ˚C; (**e**) TFA, 1:1 MeOH H2O; (**f**) PPh$_3$, Morpholine, 2,2-DPS, 0.2 M NaI in ACN; (**g**) 5-Bn-NMN, MgSO$_4$, 0.2 M MnCl$_2$ in dry formamide.

**Appendix 1—chemical structure 1.** 2-iodo-adenosine-acetonide (1). Commercially available 2-iodo-adenosine (500 mg, 1.3 mmol) was taken up into 13 ml of acetone. To this mixture, *p*-toluene sulfonic acid (three eq., 3.9 mmol) of was added and stirred for 1 hr. Consumption of the starting material was observed by TLC. The reaction was neutralized with 1 M sodium carbonate until pH = 7.0 was achieved. The product was extracted 3X with chloroform and concentrated *in vacuo* to yield a fine white crystalline powder. 70% yield. [1]H NMR (400 MHz, DMSO-*d$_6$*) δ 8.28 (d, *J* = 1.4 Hz,

*Appendix 1—chemical structure 1 continued on next page*

*Appendix 1—chemical structure 1 continued*

1H), 7.76 (s, 2H), 6.06 (d, *J* = 2.7 Hz, 1H), 5.34–5.20 (m, 1H), 5.07 (t, *J* = 5.4 Hz, 1H), 4.94 (dd, *J* = 6.2, 2.8 Hz, 1H), 4.25–4.13 (m, 1H), 3.54 (s, 2H), 1.55 (s, 3H), 1.34 (s, 3H).

**Appendix 1—chemical structure 2.** 2-ethynyl-TMS-adenosine-acetonide (2). In a clean, oven dried round bottom flask with stir bar, **1** (342 mg, 0.78 mmol) and copper iodide (0.2 eq., 0.158 mmol) were dissolved in 5 ml dry DMF. The flask was evacuated under high vacuum and refilled with argon several times. Under argon, the flask was capped with a septum and argon was allowed to bubble through a needle through the solution for 20 min. 0.1 equivalents (55 mg, 0.16 mmol) of Bis (triphenylphosphine)palladium(II) Dichloride was added to the solution while still under argon. To the capped solution, trimethylsilylacetylene (1.5, eq., 164 μl, 1.18 mmol) of was added via syringe. To this solution, triethylamine was added (three eq., 330 μl, 2.37 mmol) via syringe. The reaction was covered in aluminum foil and a balloon filled with argon was placed into the septum of the capped reaction and the reaction was allowed to progress, stirring overnight at RT. The reaction turned a deep brown color overnight. DMF was evaporated *in vacuo* in a water bath set to 40 ˚C. The residue was taken up into 25 ml ethyl acetate and remaining DMF was extracted from the organic layer with 25 ml brine (3X). The organic layer was dried over sodium sulfate, concentrated and purified via combi flash (MP A: hexanes, MP B: ethyl acetate; 0–3 min: 0–100%B, 3–18 min: 100%B). Fractions containing desired compound were pooled and concentrated to yield 191 mg of product 60% yield. [1]H NMR (400 MHz, DMSO-$d_6$) δ 8.44 (s, 1H), 7.55 (s, 2H), 6.08 (d, *J* = 2.9 Hz, 1H), 5.28 (dd, *J* = 6.1, 2.9 Hz, 1H), 5.20–5.08 (m, 1H), 4.95 (dd, *J* = 6.1, 2.7 Hz, 1H), 4.22 (s, 1H), 3.54 (t, *J* = 5.1 Hz, 2H), 1.56 (s, 3H), 1.34 (s, 3H), 0.25 (d, *J* = 1.0 Hz, 9H).

**Appendix 1—chemical structure 3.** 2-ethynyl-adenosine-acetonide (3). In a clean, dry round bottom flask, 20 ml of saturated 7N ammonia in methanol was added to **2** (200 mg, 0.5 mmol) and stirred for 1.5 hr until complete trimethylsilyl deprotection was observed by TLC. The reaction was concentrated *in vacuo*. The crude residue was purified via combi flash (MP A: hexanes, MP B: ethyl acetate; 0–1 min: 0–100% B, 1–18 min: 100% B). Fractions containing desired compound were pooled and concentrated to yield 122 mg of off-white powder (74% yield). [1]H NMR (400 MHz, DMSO-$d_6$) δ 8.43 (s, 1H), 7.55 (s, 2H), 6.10 (d, *J* = 2.9 Hz, 1H), 5.30 (dd, *J* = 6.1, 3.1 Hz, 1H), 5.16 (t, *J* = 5.5 Hz, 1H), 4.96 (dd, *J* = 6.2, 2.5 Hz, 1H), 4.23 (d, *J* = 3.0 Hz, 1H), 3.55 (q, *J* = 5.0 Hz, 2H), 1.56 (s, 3H), 1.34 (s, 3H).

**4**

**Appendix 1—chemical structure 4.** 2-ethynyl-AMP (4). In a clean, dry round bottom flask, 5.1 ml dry THF was added to **3** (0.51 mmol, 170 mg). Triethylamine (12 eq., 858 µl, 6.1 mmol) were added to the solution and cooled to 0 °C in an ice bath. Once cooled, Phosphorous (V) oxychloride (POCl$_3$) (two eq., 95 µl, 1.02 mmol,) was added dropwise to the solution. Consumption of the starting material was monitored by TLC (100% ethyl acetate). To quench the reaction, 5.1 ml of H$_2$O was added to the solution. The solution was concentrated to dryness and co-evaporated with acetonitrile and methanol. The residue was rinsed with ether. The resulting residue was resuspended in chloroform. The desired compound, insoluble in chloroform, crashed out and the resulting TEA salt was filtered out from the mixture. Without further purification the solid was dissolved in 8.2 ml of a 1:1 mixture of H$_2$O and methanol. Trifluoracetic acid (65 eq., 2.0 ml, 27.0 mmol,) was added dropwise to the solution at room temperature and stirred until complete deprotection of the acetonide moiety was observed by analytical HPLC. The mixture was concentrated to a small volume (less than 2 ml) *in vacuo* and the liquid was purified on reverse phase chromatography using a Combiflash Companion system (C18Aq 5.5 g Redisep Rf; MP A: 10 mM tributylamine/30 mM acetic acid pH 4.4 (aq.), MP B: methanol; 0–1 min: 0% B, 1–12 min: 0–50% B, 12–16 min: 100% B). Fractions containing desired product were pooled and concentrated in vacuo to yield the tributylammonium (TBA) salt of the product (80 mg, 35% yield over two steps).[1]H NMR (400 MHz, D$_2$O) δ 8.67 (s, 1H), 6.14 (d, *J* = 4.9 Hz, 1H), 4.49 (t, *J* = 4.7 Hz, 1H), 4.39 (d, *J* = 3.0 Hz, 1H), 4.24–4.11 (m, 1H), 4.06 (s, 1H), 3.36–3.24 (m, 2H).

**5**

**Appendix 1—chemical structure 5.** 2-ethynyl-AMP-morpholidate (5). In a clean oven-dried round bottom flask with stir bar, **4** (80 mg, 0.144 mmol) was taken up into 2.5 mL of DMSO (dried over 3 Å sieves overnight). 5 ml of dry DMF was co-evaporated three times *in vacuo* with the solution in a water bath set to 35 °C. After the third co-evaporation, the solution was allowed to sit under high vacuum pressure for 30 min. The solution was taken off of vacuum pressure under argon gas and capped with a septum and argon balloon. Under argon pressure and in the order listed, triphenyl
*Appendix 1—chemical structure 5 continued on next page*

*Appendix 1—chemical structure 5 continued*

phosphine (202 mg, 0.77 mmol), morpholine (108 µl, 1.24 mmol), and 2,2'-dipyridyl disulfide (170 mg, 0.77 mmol) were added to the solution. The reaction was allowed to take place for 1.5 hr until consumption of the starting material was observed by HPLC. The desired compound was precipitated out in 0.2 M NaI in acetonitrile (dry) and filtered using a fine filter glass funnel. The result was an off-white powder (36 mg, 57% yield). $^1$H NMR (400 MHz, D$_2$O) δ 8.45 (s, 1H), 6.04 (d, $J$ = 4.8 Hz, 1H), 4.49 (t, $J$ = 4.7 Hz, 1H), 4.31 (s, 1H), 3.99 (d, $J$ = 12.0 Hz, 2H), 3.51 (s, 4H), 3.30 (s, 1H), 2.98–2.73 (m, 4H).

**Appendix 1—chemical structure 6.** 5-benzyl-2-ethynyl-NAD$^+$ (6). 5-benzyl-nicotinamide mononucleotide (two eq., 29 mg, 0.068 mmol), synthesized as previously reported (*Carter-O'Connell et al., 2016*), was taken up into 2 ml dry toluene in a scintillation vile and concentrated *in vacuo* (3X) to remove excess water. 30 min under high vacuum, the solid was capped with a septum under argon pressure. **5** (15 mg, 0.034 mmol), after overnight desiccation in a chamber containing P$_2$O$_5$, was added along with MgSO$_4$ (8 mg, 0.068 mmol) to the scintillation vile containing 5-Bn-NMN under argon pressure. 500 µl of 0.2 M MnCl$_2$ in dry formamide was added to the solids via syringe and the reaction was stirred overnight under Argon balloon at room temperature. The reaction was checked by HPLC. After complete consumption of **5** was observed overnight, the reaction was diluted with 1.5 mL 10 mM tributylamine/30 mM acetic acid [pH = 4.4] and purified using HPLC. Retention time 8.5 min. The fractions containing desired compound were pooled and co-evaporated with acetonitrile (to remove excess water) and methanol to remove excess tributylamine. The resulting compound was a white powder (13 mg, 33% yield). $^1$H NMR (400 MHz, D$_2$O) δ 9.16 (s, 1H), 9.00 (s, 1H), 8.87 (s, 1H), 8.60 (s, 1H), 8.52 (s, 1H), 7.29 (m, 5H), 5.98 (d, J = 5.0 Hz, 1H), 5.92 (d, J = 5.2 Hz, 1H), 4.30–4.24 (m, 11H), 3.53 (s, 1H).

## 1H NMR spectra

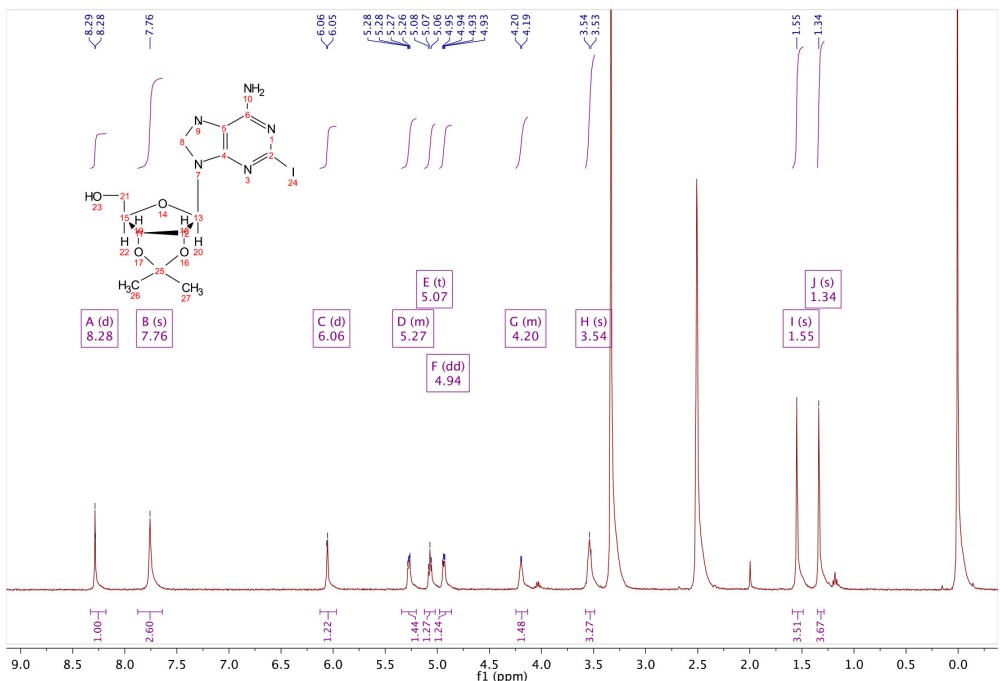

**Appendix 1—figure 1.** [1]H NMR of compound 1.

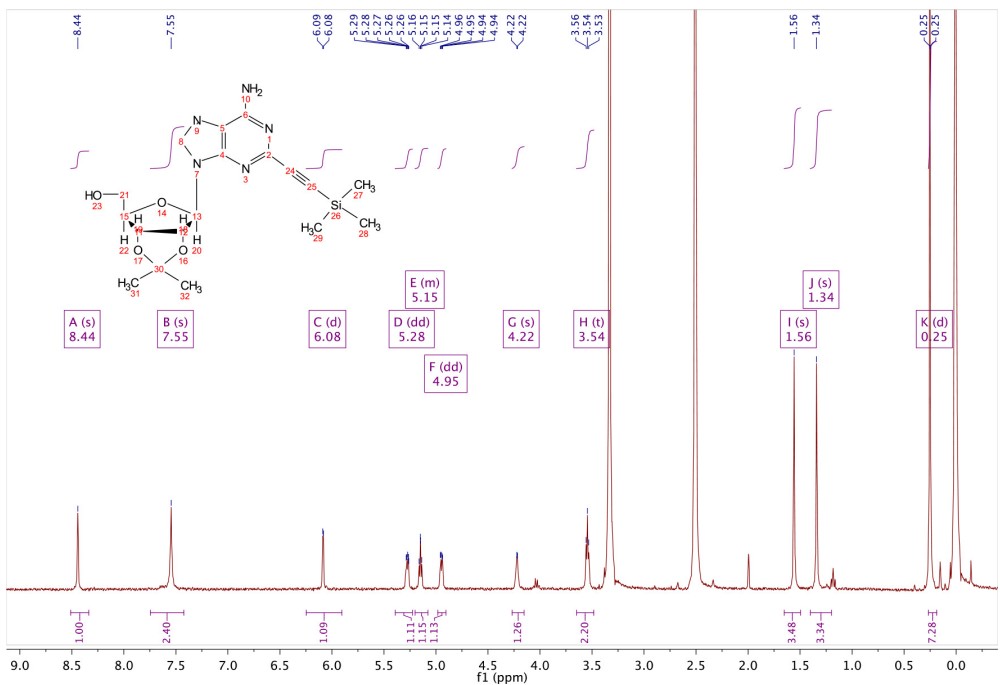

**Appendix 1—figure 2.** [1]H NMR of compound 2.

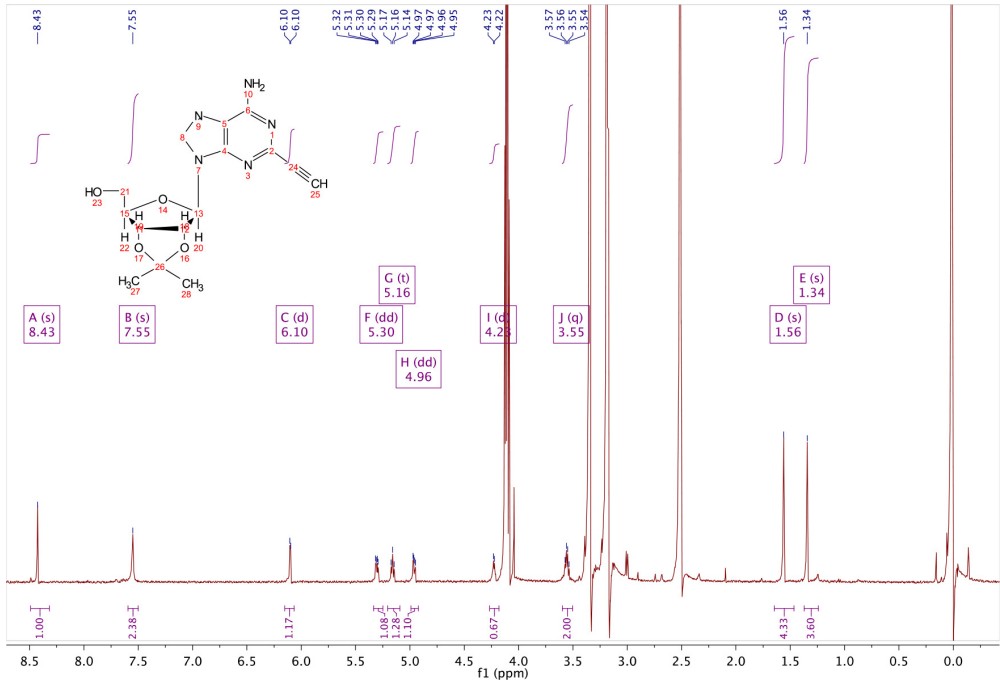

**Appendix 1—figure 3.** $^1$H NMR of compound 3.

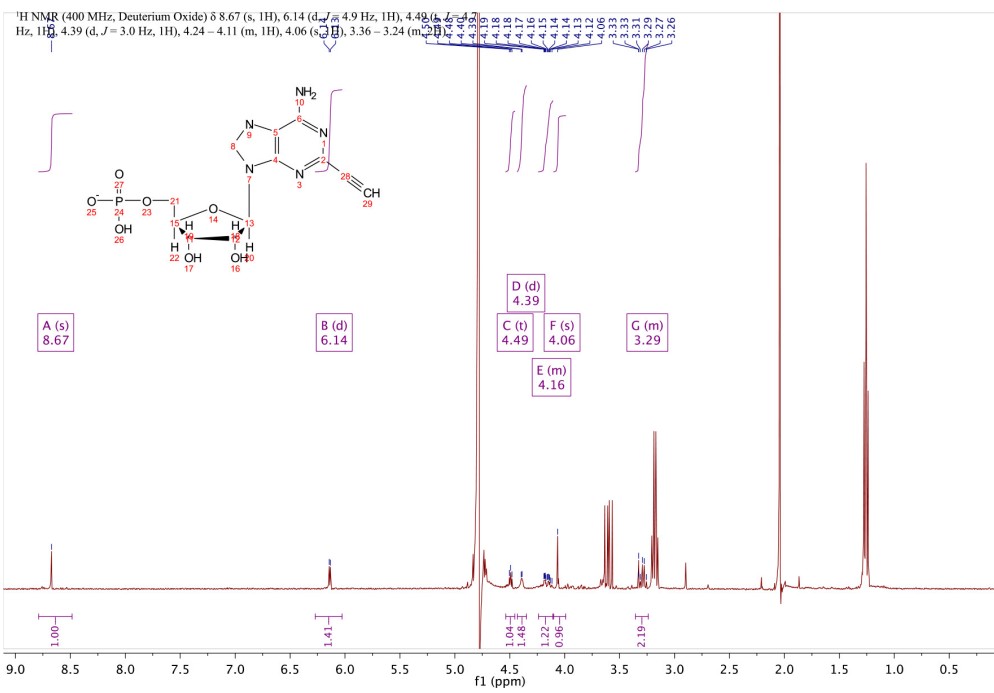

**Appendix 1—figure 4.** $^1$H NMR of compound 4.

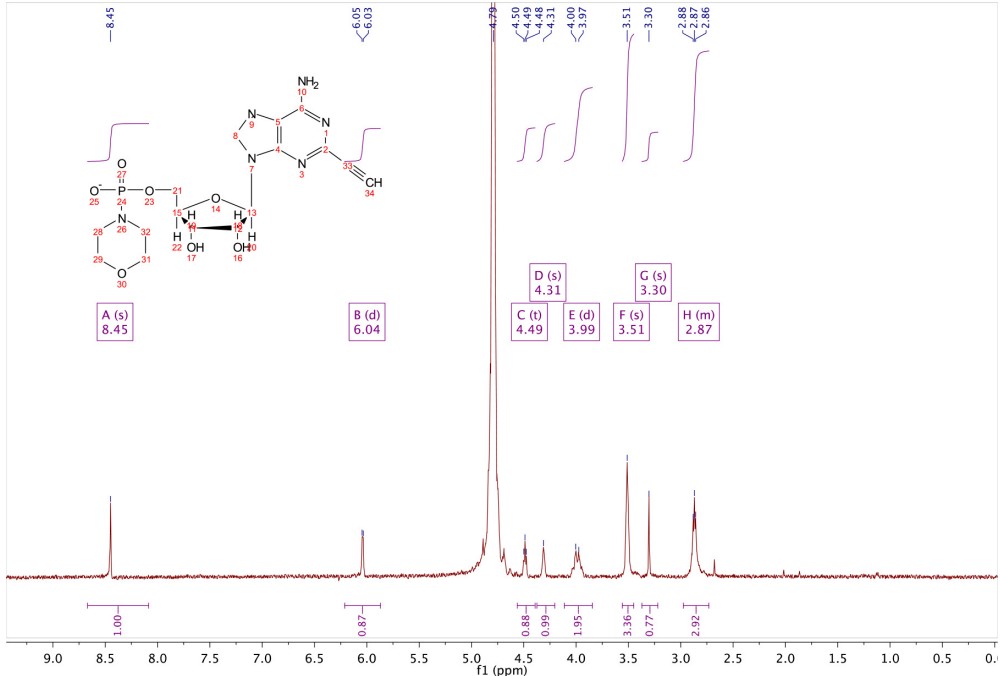

**Appendix 1—figure 5.** ʲ$^1$H NMR of compound 5.

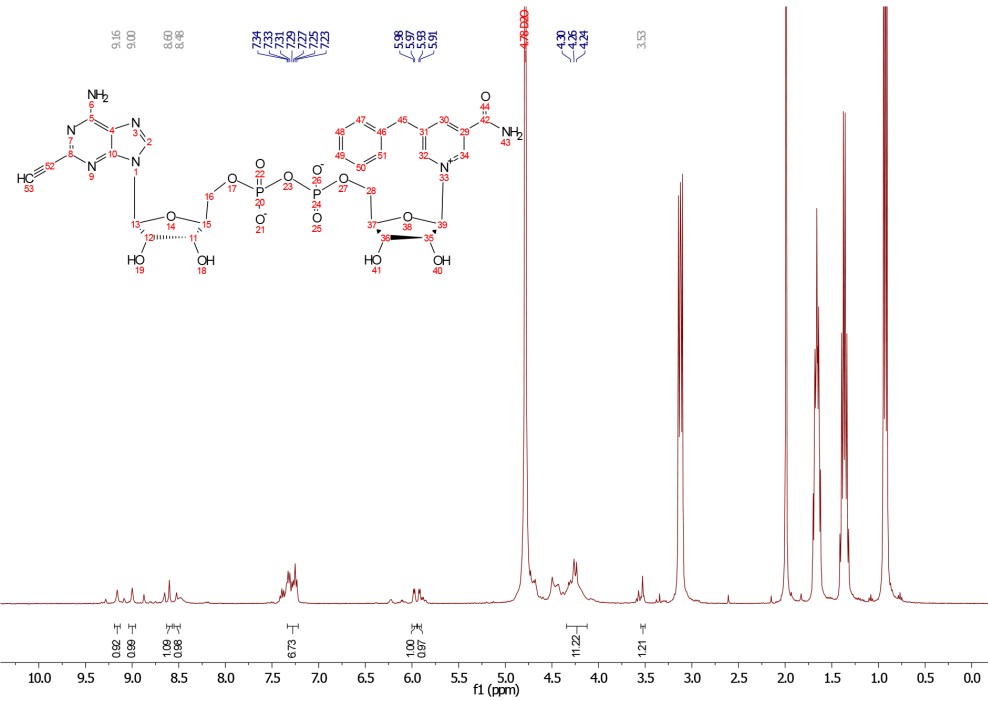

**Appendix 1—figure 6.** $^1$H NMR of compound 6 (5-benzyl-2-ethynyl-NAD$^+$).

## Mass spectrometry 5-bn-2e-NAD+: exact mass 776.15

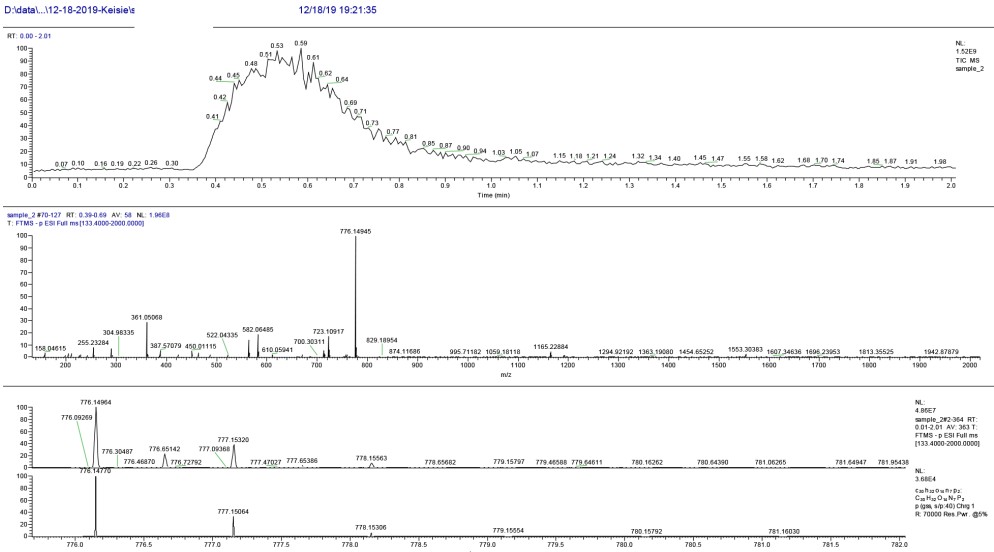

**Appendix 1—figure 7.** MS data for 5-benzyl-2-ethynyl-NAD$^+$.

**Appendix 1—chemical structure 7.** 2-fluoro-5-((4-oxo-3,4-dihydrophthalazin-1-yl)methyl)benzamide. 2-fluoro-5-((4-oxo-3,4-dihydrophthalazin-1-yl)methyl)benzoic acid (50 mg, 0.17 mmol) was added to a flame dried flask and dissolved in SOCl$_2$ (20 ml) and refluxed at 80 °C for 45 min. The reaction mixture was cooled to 0 °C and then added dropwise to ice cold anhydrous NH$_3$/MeOH (7 N, 100 ml). White solids quickly precipitated from an orange solution accompanied by rapid heating of the reaction vessel and smoke. The reaction was left to stir over night and the product collected as a solid by vacuum filtration: 50 mg (100%).

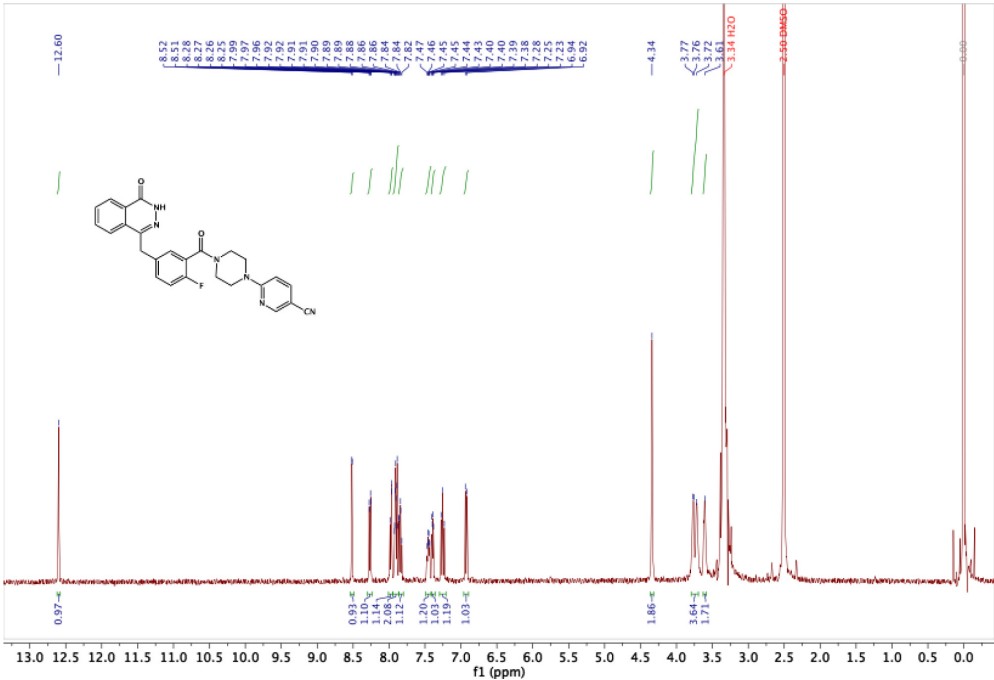

**Appendix 1—chemical structure 8.** 6-(4-(2-fluoro-5-((4-oxo-3,4-dihydrophthalazin-1-yl)methyl)ben-zoyl)piperazin-1-yl)nicotinonitrile (Phthal 01). 2-fluoro-5-((4-oxo-3,4-dihydrophthalazin-1-yl)methyl) benzoic acid (50 mg, 0.17 mmol), TBTU (60 mg, 0.19 mmol) and 6-(piperazin-1-yl)nicotinonitrile (36 mg, 0.19 mmol) were added to a flame dried flask and dissolved in anhydrous DMF/DCM (1:10) (5 mL). DIPEA (0.06 mL, 0.34 mmol) was added under argon and the reaction stirred at RT for 18 h. The product was then precipitated in water (100 mL) and collected by vacuum filtration as a cream solid: 70 mg (89%). $^1$H NMR (400 MHz, DMSO-$d_6$) δ 12.60 (s, 1H), 8.51 (d, J = 2.3 Hz, 1H), 8.27 (dd, J = 7.7, 1.5 Hz, 1H), 8.01 – 7.95 (m, 1H), 7.95 – 7.87 (m, 2H), 7.87 – 7.80 (m, 1H), 7.45 (td, J = 5.5, 5.0, 2.7 Hz, 1H), 7.39 (dd, J = 6.5, 2.3 Hz, 1H), 7.25 (t, J = 9.0 Hz, 1H), 6.93 (d, J = 9.1 Hz, 1H), 4.34 (s, 2H), 3.79 – 3.69 (m, 4H), 3.61 (s, 2H). MS m/z [M-H]- for $C_{26}H_{21}FN_6O_2$: 466.7. $R_t$ = 8.95 min.

## NMR Spectra: Phthal 01

**Appendix 1—figure 8.** $^1$H NMR of Phthal01.

## Mass spectrometry- Phthal 01-exact mass 466.7

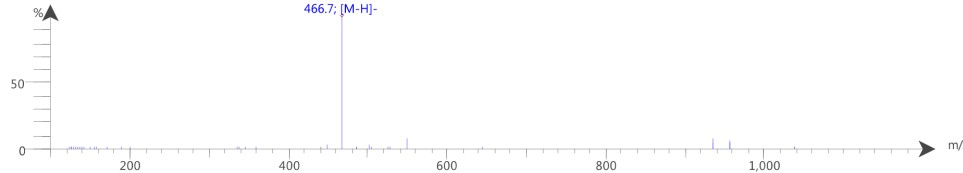

**Appendix 1—figure 9.** MS data for Phthal01.

