## [Decision Letter]

**Acceptance summary:**

Using chemical genetics approaches, this study along with the back-to-back study from the Kraus group represent the first systematic investigations on PARP-7 regulated proteome. Particularly, this study use tour-de-force approaches (chemical genetics, proximity ligation, inhibitor) to analyze PARP-7 regulated proteome and identify the specificity of PARP-7 on cysteine residues at a proteomics level using mass spectrometry approaches. Given the recent findings on the role of PARP-7 in viral infection and the development of PARP-7 inhibitor for cancer therapy, this work is timely for the field and beyond.

**Decision letter after peer review:**

Thank you for submitting your article "Chemical genetics and proteome-wide site mapping reveal cysteine MARylation by PARP-7 on immune-relevant protein targets" for consideration by *eLife*. Your article has been reviewed by three peer reviewers, including Anthony K L Leung as the Reviewing Editor and Reviewer #1, and the evaluation has been overseen by Philip Cole as the Senior Editor.

The reviewers have discussed the reviews with one another and the Reviewing Editor has drafted this decision to help you prepare a revised submission.

As the editors have judged that your manuscript is of interest, but as described below that additional experiments are required before it is published, we would like to draw your attention to changes in our revision policy that we have made in response to COVID-19 (https://elifesciences.org/articles/57162). First, because many researchers have temporarily lost access to the labs, we will give authors as much time as they need to submit revised manuscripts. We are also offering, if you choose, to post the manuscript to bioRxiv (if it is not already there) along with this decision letter and a formal designation that the manuscript is "in revision at *eLife*". Please let us know if you would like to pursue this option.

Summary:

Rodriguez et al. report a thorough investigation of PARP7 activity by refining and employing several orthogonal methods to identify interactors and enzymatic targets. The authors have created a new variant of the chemical genetics approach for identifying PARP7, which will also be potentially useful for PARP12 (although data not shown). In addition, they used a proximity ligation (BioID) method to identify PARP7 interactors. Using these two methods, they identified PARP13 as a high-ranking target, which was then confirmed by (1) a new inhibitor (Phthal01) that has a preference for PARP7, and (2) identifying the ADP-ribosylation site of PARP13 upon overexpressing of PARP7. Lastly, the proteomics analyses also provided further confirmation of the specificity of PARP7 towards cysteine, which was indirectly inferred by the sensitivity of HgCl2 prior to this study. The study was well-executed and provides an important and comprehensive resource of PARP7 substrates and ADP-ribosylation sites. Given the growing interest in PARP7 as an important player in the antiviral immune response, this study is timely and relevant. However, confidence in the biological relevance of the presented results could be significantly expanded with a few key experiments.

Essential revisions:

1) Biological role of Cys-ADP-ribosylation of PARP13: This study has a lot of technical strengths. However, it is unclear about the biological effect of PARP7-mediated Cys MARylation on PARP13. For example, given the zinc finger is critical for RNA-binding, would the MARylation abrogate the RNA-binding ability? in vitro purified proteins would allow the authors to test the effect of PARP7-mediated MARylation on PARP13 RNA binding, which is important for viral response.

2) Endogenous interaction between PARP13 and PARP7: Another limitation of the current study is that the identification of the PARP13 as PARP-7 target was mainly through overexpression study. Therefore, it is critical for the authors to demonstrate that the modification on PARP13 is due to endogenous interaction and catalytic activity.

Do endogenous PARP13 and PARP7 physically interact in untreated, or virally-challenged, WT cells, and is PARP13 ADP-ribosylated in this context? Would this interaction hold true for both isoforms? E.g., Pulldown of endogenous PARP7/PARP13 followed by western blot with PARP13/PARP7, pulldown of PARP13 followed by western blot with anti-PAN antibody, and pulldown of PAN-ADPr proteins followed by anti-PARP13 western blot in the context of WT and Pthal01 treated cells would gain confidence in the manuscript's overall conclusions. Endogenous ADP-ribosylation of PARP13 cannot be detected according to Figure 3B (only upon PARP7 overexpression). A pull-down of PARP13 as a way of enriching the protein could be used to test whether PARP13 shows endogenous ADP-ribosylation in a PARP-7 dependent manner.

Can the author exclude the possibility that the ADP-ribosylation of PARP13 is an indirect effect of the expression instead of its catalytic activity? Would the PARP13 ADP-ribosylation occur in the presence of PARP7 catalytically dead mutant?

Can the author comment on whether PARP13 is a bona fide, endogenous PARP7-target during viral infections of WT cells? What would the authors predict how the cells respond to viral infections upon PARP7 overexpression or treatment with Pthal01?

3) Half-life analyses: It was estimated that the half-life of PARP-10 is ~15 min and that of PARP13.2 is 189 min. Based on these analyses, the authors inferred that Cys-ADPr bond is more stable in cells than Glu/Asp-ADPr-bond. However, the analyses were performed at different concentrations of Phthal01. Would the higher concentration of Phthal01 affect the half-life measured on PARP13.2?

Moreover, could the observed loss of signal not only be dependent on the rate of removal, but may also be influenced by the amount of the respective PARPs ADPr at the start of Phthal01 treatment. If, under these settings, PARP7-mediated ADPr is much more abundant than PARP10-mediated ADPr, PARP7 ADPr could be longer-lived while having the same removal kinetics as PARP10 ADPr. Given that the authors are making a claim on the underlying kinetics this possible confounder should be addressed.

Lastly, the authors should clarify whether the difference in stability is due to the chemical or enzymatic stability of the two ADPr bonds. To measure the chemical stability, the authors could, for example, lyse the cells in denaturing conditions by keeping the pH at physiological levels and monitor the ADPr levels at different times similarly to the experiment in Figure 4D-E.

4) ZnF Cys mutations on protein stability: A couple of data regarding PARP13 stability need clarification: a) Figure 4D: Treatment with Phthal01 not only reduces PARP7 and PARP13.2 MARylation but also increases their protein levels, suggesting that MARylation has a negative effect on protein stability. b) Figure 5D: PARP7 overexpression reduces PARP13.2 protein levels. Cys to Ala mutations in PARP13 ZnF also reduce PARP13.2 protein levels compared to WT, especially when all four Cys were mutated. Reduction in Cys MARylation upon Cys mutation can be due to reduction in protein stability. Cys are structural elements of ZnF clusters and their mutation (to Ala) leads to loss of protein stability. Given that Cys within Zn fingers of PARP13 are MARylated, the modification could also lead to disruption of Zn clusters and partial (or complete) denaturation. This possibility could be tested with in vitro purified proteins (PARP7 and PARP13) using CD spectroscopy to validate the effect of Cys mutations and PARP7-mediated MARylation on PARP13 secondary structure.

---

## [Author Response]

Essential revisions:1) Biological role of Cys-ADP-ribosylation of PARP13: This study has a lot of technical strengths. However, it is unclear about the biological effect of PARP7-mediated Cys MARylation on PARP13. For example, given the zinc finger is critical for RNA-binding, would the MARylation abrogate the RNA-binding ability? in vitro purified proteins would allow the authors to test the effect of PARP7-mediated MARylation on PARP13 RNA binding, which is important for viral response.

We thank the reviewers for their comments regarding the technical strengths of our study. We agree that understanding the biological function for PARP-7 mediated Cys MARylation of PARP-13 is important. We asked if PARP-7 mediated MARylation of PARP-13 impacts the ability of PARP-13 to bind to a host mRNA, an experiment that we could perform in our lab. This experiment is based on a recent manuscript that showed that PARP-13.2 (ZAP-S) binds to Interferon Λ 3 (IFNL3) and mediates its degradation (Schwerk et al., 2019). In their manuscript, they developed a biotinylated RNA probe for Interferon Λ 3 (IFNL3) and incubated this probe with mammalian cell lysates. Using tandem mass spectrometry, they identified PARP-13.2 as an interacting protein by mass spectrometry.

We expressed PARP-7 in HEK 293T cells (conditions in which we found that PARP-7 MARylates endogenous PARP-7 on Cys—predominately on zinc finger Cys—by mass spectrometry) and treated with either Phthal01 (to inhibit PARP-7) or veliparib (PARP-1/2 inhibitor; control since Phthal01 inhibits PARP-1/2 at the tested concentration) overnight. We harvested cells and incubated lysates derived from the aforementioned cell-treatment conditions with the biotinylated IFNL3 RNA probe followed by streptavidin enrichment. While we did find that PARP-13 isoforms were enriched with the biotinylated IFNL3 RNA probe, expression of PARP-7 nor inhibition of its catalytic activity affected PARP-13 binding to IFNL3 RNA (Author response image 1, two replicates). Similar results were obtained when we co-expressed PARP-7 with PARP-13-2 (data not shown).

These results suggest that PARP-7 mediated MARylation of PARP-13 isoforms doesn’t affect their ability to bind IFNL3 RNA. One caveat is that it is possible that there is low stoichiometry of PARP-13 MARylation by PARP-7 under these conditions (unfortunately we do not have a way to quantitively assess MARylation of PARP-13). Another possibility is that IFNL3 RNA does not bind to the N-terminal zinc finger but to a different region of PARP-13 in the Schwerk et al. paper they did not explicitly show that IFNL3 RNA binds to the N-terminal zinc finger domain. Lastly, it is possible that context (e.g. viral infection) is important for the ability of PARP-7 to regulate PARP-13 binding to RNA, perhaps due to other interaction partners or other post-translational modifications. Unfortunately, we do not have the ability to evaluate the effects of PARP-7 on PARP-13 binding to RNA during viral infection; however, in future studies we plan to examine this with collaborators. We also will determine if perhaps PARP-7 disrupts viral RNA binding but not host RNA binding to PARP-13. For these experiments we will need to develop and validate biotinylated viral RNA probes, which we feel is beyond the scope of the current study.

2) Endogenous interaction between PARP13 and PARP7: Another limitation of the current study is that the identification of the PARP13 as PARP-7 target was mainly through overexpression study. Therefore, it is critical for the authors to demonstrate that the modification on PARP13 is due to endogenous interaction and catalytic activity.Do endogenous PARP13 and PARP7 physically interact in untreated, or virally-challenged, WT cells, and is PARP13 ADP-ribosylated in this context? Would this interaction hold true for both isoforms? E.g., Pulldown of endogenous PARP7/PARP13 followed by western blot with PARP13/PARP7, pulldown of PARP13 followed by western blot with anti-PAN antibody, and pulldown of PAN-ADPr proteins followed by anti-PARP13 western blot in the context of WT and Pthal01 treated cells would gain confidence in the manuscript's overall conclusions. Endogenous ADP-ribosylation of PARP13 cannot be detected according to Figure 3B (only upon PARP7 overexpression). A pull-down of PARP13 as a way of enriching the protein could be used to test whether PARP13 shows endogenous ADP-ribosylation in a PARP-7 dependent manner.

The reviewers bring up good points. To demonstrate that endogenous PARP-7 and PARP-13 interact in cells, we attempted co-IP experiments. For these experiments we collaborated with Dr Jason Matthews, who has developed a PARP-7 specific antibody (validated in Figure 4—figure supplement 2 using PARP-7 KO mouse embryonic fibroblasts; comparison with commercially available PARP-7 antibody is shown). Unfortunately, the expression of PARP-7 under basal conditions was undetectable in HEK 293T or A549 cells. Therefore we could not perform co-IP experiments looking at endogenous PARP-7— PARP-13 interactions. It is possible that viral infection will induce PARP-7 expression (and activity) to a level that we can detect by Western blot, and in future studies with collaborators we will explore these conditions for co-IP experiments.

To further substantiate PARP-13 as a substrate of PARP-7 we expressed (in *E. coli*) and purified GST-tagged full-length PARP-7 and full-length His_6_-SUMO-PARP-13.2. We found that PARP-7 trans-MARylates PARP-13.2 (in addition to auto-MARylation) in vitro in a time dependent manner (Figure 3—figure supplement 1B). By contrast full-length His_6_-SUMO PARP-10 strongly auto-MARylates but only weakly trans-MARylates PARP-13.2 (Figure 3—figure supplement 1B). This is consistent with our cell-based experiments in which GFP-PARP-7, but not GFP-PARP10, trans-MARylates PARP-13.2 (Figure 4). Together these results further support the notion that PARP-13.2 is a bona fide target of PARP-7.

Having purified full-length, recombinant PARP-7 and PARP-13.2, we determined if (i) they interact in vitro and (ii) if the interaction is dependent on PARP-7 catalytic activity. Consistent with the BioID studies, we found that GST-PARP-7 and His_6_-SUMO-PARP-13.2 interact in vitro as determined by GST-pulldown experiments. Neither addition of NAD^+^ nor inhibition of PARP-7 catalytic activity with Phthal01 altered the interaction between PARP-7 and PARP-13.2 (Figure 3—figure supplement 1C). These results show that the interaction between PARP-7 and PARP-13.2 is independent of PARP-7 catalytic activity.

Can the author exclude the possibility that the ADP-ribosylation of PARP13 is an indirect effect of the expression instead of its catalytic activity? Would the PARP13 ADP-ribosylation occur in the presence of PARP7 catalytically dead mutant?

We believe that our results showing that Phthal01 (1 μM; inhibits PARP-7, PARP-1, and PARP-2 at this concentration in cells), but not veliparib (1 μM; inhibits PARP-1 and PARP-2 at this concentration in cells) inhibits PARP-7 mediated trans-MARylation of PARP-13.2 (Figure 4—figure supplement 1F) in cells strongly supports the notion that the catalytic activity of PARP-7 is required for PARP-7-meidated MARylation of PARP-13.2 in cells.

Can the author comment on whether PARP13 is a bona fide, endogenous PARP7-target during viral infections of WT cells? What would the authors predict how the cells respond to viral infections upon PARP7 overexpression or treatment with Pthal01?

These are great questions. As stated above, in future studies with virology lab collaborators we will examine whether PARP-13.1 and/or PARP-13.2 are trans-MARylation targets of PARP-7 during viral infection. Intriguingly, PARP-7 appears to play an anti-viral against some viruses (e.g. Sindbis virus) (Kozaki et al., 2017), and a pro-viral against other viruses (e.g. influenza A virus) (Yamada et al., 2016). Additionally, a recent showed that PARP-13.1 and PARP-13.2 have distinct roles in antiviral innate immunity: PARP-13.1 targets viral (e.g. Sindbis) RNA and inhibits viral replication, whereas PARP-13.2 targets host interferon RNA and suppresses interferon signaling degradation (Schwerk et al., 2019). Given the varied responses of PARP-7 to viral infection and the distinct roles for PARP-13.1 and PARP-13.2 during viral infection, it is difficult to know how cells will respond to viral infection in the context of PARP-7 overexpression or inhibition of its catalytic activity. Since PARP-13.1 and PARP-13.2 are both trans-MARylation targets of PARP-7, it is possible that PARP-7 could regulate both PARP-13.1 and PARP-13.2 during viral infection (which might also depend on virus type).

3) Half-life analyses: It was estimated that the half-life of PARP-10 is ~15 min and that of PARP13.2 is 189 min. Based on these analyses, the authors inferred that Cys-ADPr bond is more stable in cells than Glu/Asp-ADPr-bond. However, the analyses were performed at different concentrations of Phthal01. Would the higher concentration of Phthal01 affect the half-life measured on PARP13.2?

We do not think higher concentrations of Phthal01 would impact PARP-13.2 MARylation half-life measurements since 1μM Phthal01 was sufficient to completely inhibit PARP-7 in cells. Ideally, we would perform the experiment using a structurally distinct PARP-7 inhibitor. We are hopeful that one will become commercially available soon.

Moreover, could the observed loss of signal not only be dependent on the rate of removal, but may also be influenced by the amount of the respective PARPs ADPr at the start of Phthal01 treatment. If, under these settings, PARP7-mediated ADPr is much more abundant than PARP10-mediated ADPr, PARP7 ADPr could be longer-lived while having the same removal kinetics as PARP10 ADPr. Given that the authors are making a claim on the underlying kinetics this possible confounder should be addressed.

The reviewer raises an important consideration. We do not believe this is a concern, however, since the signal for PARP-7 mediated PARP-13.2 MARylation is similar to or lower than the signal for PARP10 MARylation (see for example Figure 4B).

Lastly, the authors should clarify whether the difference in stability is due to the chemical or enzymatic stability of the two ADPr bonds. To measure the chemical stability, the authors could, for example, lyse the cells in denaturing conditions by keeping the pH at physiological levels and monitor the ADPr levels at different times similarly to the experiment in Figure 4D-E.

The reviewer raises an important issue that needs to be addressed. We prepared lysates under denaturing conditions (2% SDS, 95°C, pH 7.4) from HEK 293T cells expressing either PARP-10 or PARP-7 together with PARP-13.2. Samples were allowed to sit at room temperature and aliquots were removed at different time points and proteins were resolved by SDS-PAGE and detected Western blot using antibodies against ADPr, GFP, and actin. We find that the MARylated PARP-10, PARP-7, and PARP-13.2 are stable for at least 24 hours (Figure 4—figure supplement 2A). This results supports to the notion that the differences in the cellular half-life for MARylated PARP-10 versus MARylated PARP-13.2 reflect differences in the enzymatic stability of Glu/Asp-ADPr versus Cys-ADPr.

4) ZnF Cys mutations on protein stability: A couple of data regarding PARP13 stability need clarification: a) Figure 4D: Treatment with Phthal01 not only reduces PARP7 and PARP13.2 MARylation but also increases their protein levels, suggesting that MARylation has a negative effect on protein stability.

We agree with the reviewer that this is an interesting observation. We sought to further examine the effects of Phthal01 on endogenous PARP-7 and PARP-13. Using the novel PARP-7 specific antibody (described above), we found that Phthal01 substantially increased the levels of endogenous PARP-7 in A549 cells and MEFs (Figure 4—figure supplement 2B), consistent with results obtained using GFP-PARP-7. This is an interesting finding and suggests that PARP-7 catalytic activity regulates its stability. Under these conditions, however the levels of endogenous PARP-13.1 or PARP-13.2 did not change (Figure 4—figure supplement 2C). This suggest that PARP-7 mediated MARylation of PARP-13 does not impact its stability; however, it is also possible that the effects of PARP-7 on PARP-13 are context dependent (for example, during viral infection). We will explore the latter possibility in future studies. We will also explore the mechanism by which Phthal01 stabilizes PARP-7.

b) Figure 5D: PARP7 overexpression reduces PARP13.2 protein levels. Cys to Ala mutations in PARP13 ZnF also reduce PARP13.2 protein levels compared to WT, especially when all four Cys were mutated. Reduction in Cys MARylation upon Cys mutation can be due to reduction in protein stability. Cys are structural elements of ZnF clusters and their mutation (to Ala) leads to loss of protein stability. Given that Cys within Zn fingers of PARP13 are MARylated, the modification could also lead to disruption of Zn clusters and partial (or complete) denaturation. This possibility could be tested with in vitro purified proteins (PARP7 and PARP13) using CD spectroscopy to validate the effect of Cys mutations and PARP7-mediated MARylation on PARP13 secondary structure.

This is a great point raised by the reviewer and something that we’ve contemplated. We do not have access to CD spectroscopy, but we could collaborate with a colleague in the future. Additionally, we need to optimize the expression and purification of bacterially expressed PARP-13.2 to obtain enough PARP-13.2 of sufficient purity for CD spectroscopy experiments. Lastly, we need to optimize a protocol for isolating unmodified versus MARylated PARP-13.2.